**\*For correspondence:**
Anna.Norrby-Teglund@ki.se
(AN-T);
us6@st-andrews.ac.uk (US-L)

†These authors contributed equally to this work
‡These authors also contributed equally to this work

**Present address:** §Department of Biochemistry, University of Cambridge, Cambridge, United Kingdom; #Department of Microbiology, Faculty of Pharmacy, Meijo University, Nagoya, Japan; ¶Evotec Ltd, Abingdon, United Kingdom; \*\*Howard Hughes Medical Institute and Department of Biology, Brandeis University, Waltham, United States; ††Laboratory of Tissue Biology and Therapeutic Engineering, CNRS UMR 5305, University Claude Bernard-Lyon 1, Lyon, France; ‡‡University of Lyon, Lyon, France; §§Triple Helical Peptides Ltd, Cambridge, United Kingdom

**Competing interest:** The authors declare that no competing interests exist.

# Structural basis for collagen recognition by the *Streptococcus pyogenes* M3 protein and its involvement in biofilm

Marta Wojnowska[1†§], Takeaki Wajima[2†#], Tamas Yelland[3¶], Hannes Ludewig[1\*\*], Robert M Hagan[1], Olivia F McCurry[1], Grant Watt[1], Samir W Hamaia[4], Dominique Bihan[4], Jean-Daniel Malcor[4††, ‡‡], Arkadiusz Bonna[4], Helena Bergsten[2], Laura Marcela Palma Medina[2], Mattias Svensson[2], Oddvar Oppegaard[5,6], Steinar Skrede[5,6], Per Arnell[7], Ole Hyldegaard[8,9], Richard W Farndale[4§§], Anna Norrby-Teglund[2\*‡], Ulrich Schwarz-Linek[1\*‡]

[1]School of Biology, Biomedical Sciences Research Complex, University of St Andrews, North Haugh, St Andrews, United Kingdom; [2]Center for Infectious Medicine, Karolinska Institutet, Karolinska University Hospital Huddinge, Stockholm, Sweden; [3]CRUK Beatson Institute, Glasgow, United Kingdom; [4]Department of Biochemistry, University of Cambridge, Downing Site, Cambridge, United Kingdom; [5]Department of Medicine, Haukeland University Hospital, Bergen, Norway; [6]Department of Clinical Science, University of Bergen, Bergen, Norway; [7]Department of Anesthesiology and Intensive Care Medicine, Sahlgrenska University Hospital, Gothenburg, Sweden; [8]Department of Anesthesiology, Hyperbaric Medicine Center, Head and Orthopedic Center, Copenhagen University Hospital, Rigshospitalet, Copenhagen, Denmark; [9]Department of Clinical Medicine, University of Copenhagen, Copenhagen, Denmark

## eLife Assessment

M proteins are essential group A streptococci virulence factors that bind to numerous human proteins; a small subset of M proteins, such as M3, have been reported to bind collagen, which is thought to promote tissue adherence. In this **important** paper, the authors provide a **solid** characterization of M3 interactions with collagen. The work raises significant questions regarding the specificity of the structure and its interactions with different collagens, with implications for the variable actions of M protein collagen interactions on biofilm formation.

**Abstract** The M protein is an essential virulence factor of *Streptococcus pyogenes*, or group A streptococcus (GAS), one of the most common and dangerous human pathogens. Molecular and functional characterization of M protein variants and their interactions with host components is crucial for understanding streptococcal pathogenesis and vaccine development. The M3 protein is produced by the prevalent *emm*3 GAS serotype, which is frequently associated with severe invasive diseases. Here, we structurally and biochemically characterize the interaction of M3 with human collagens. High-resolution structures of the N-terminal M3 domain in the free state as well as bound to a collagen peptide derived from the Collagen Ligands Collection reveal a novel T-shaped protein fold that presents binding sites complementing the characteristic topology of collagen triple helices. The structure of the M3/collagen peptide complex explains how *emm*3 GAS and related streptococci, such as *Streptococcus dysgalactiae* subsp. *equisimilis*, can target collagens to enable colonization of various tissues. In line with this, we demonstrate that the M3/collagen interaction promotes enhanced biofilm formation of *emm*3 GAS in an *emm* type-specific manner, which can be

inhibited with the recombinant N-terminal M3 domain. Further, *emm*3 GAS are shown to colocalize with collagen in tissue biopsies from patients with necrotizing soft tissue infections, where GAS biofilms are common. This observation is reproduced in infected organotypic skin models. Together, these data provide detailed molecular insights into an important streptococcal virulence mechanism with implications for the understanding of invasive infections, strategies for treating biofilm and M-protein-based vaccine design.

## Introduction

*Streptococcus pyogenes* (group A streptococcus [GAS]) is one of the most prevalent human pathogens. It causes acute diseases ranging from trivial to life-threatening, such as pharyngotonsillitis (strep throat), scarlet fever, impetigo, meningitis, necrotizing fasciitis, and streptococcal toxic shock-like syndrome (*Brouwer et al., 2023*). The highest disease burden is caused by the post-infection sequelae acute rheumatic fever, rheumatic heart disease, and glomerulonephritis (*Carapetis et al., 2005*; *Jackson et al., 2011*), but there is also a worrying global rise of severe invasive infections (*Bagcchi, 2023*). A key virulence factor involved in many, if not all, these diseases is the GAS M protein. Encoded by the chromosomal *emm* gene, the M protein is covalently anchored at its C-terminus to the cell wall, covering the GAS surface at high density and acting as an adhesin and a potent immune evasion factor (*Oehmcke et al., 2010*; *Smeesters et al., 2010*; *Walker et al., 2014*). M proteins are known, or can confidently be predicted, to form linear fibrils extending approximately 50 nm from the bacterial surface (*Phillips et al., 1981*; *Nilson et al., 1995*). Their hair-like architecture is based on dimerization of α-helical chains into parallel coiled coils, the defining and dominant structural feature of this class of proteins. While sharing similar overall structure, M proteins are sequentially highly diverse, with over 220 distinct known variants of the *emm* gene defining GAS serotypes (*Sanderson-Smith et al., 2014*). The C-terminal regions of M protein variants are highly conserved and predicted to form well-defined coiled coils. Sequences diverge increasingly toward the N-terminal hypervariable region (HVR) that is projected away from the bacterial surface. Experimental high-resolution structural information for M proteins is limited (*Macheboeuf et al., 2011*; *McNamara et al., 2008*; *Stewart et al., 2016*; *Buffalo et al., 2016*).

In addition to their role in adhesion and subverting the host's immune response, M proteins have also been implicated in biofilm formation (*Fiedler et al., 2015*). Bacterial biofilms are a major cause of difficult-to-treat infections largely contributed to by their recalcitrance to antibiotics and immune responses (*Hoiby et al., 2014*). They are classified as either surface-attached biofilms, typically associated with implants and medical device-associated infections, or non-surface-associated biofilms. The latter are commonly observed in respiratory infections of patients with impaired mucociliary function and in persistent soft tissue infections seen in chronic wounds resulting from diabetes or impaired vascularization (*Ciofu et al., 2022*). Biofilm is also a potentially complicating feature associated with severe invasive GAS diseases. Analyses of biopsies collected during the acute phase identified biofilm in over 30% of patients with necrotizing soft tissue infections caused by GAS (*Siemens et al., 2016*). Some GAS types, such as *emm*1 and *emm*3, are over-represented among isolates from severe invasive infections, that is, necrotizing soft tissue infections and streptococcal toxic shock syndrome (*Brouwer et al., 2023*; *Luca-Harari et al., 2009*; *Bruun et al., 2021*), but no clear association between biofilm formation and *emm* type is known (*Lembke et al., 2006*). Köller et al. highlighted that the propensity of GAS to form biofilm in vitro was directly dictated by the experimental setting where both the medium and coating with specific extracellular matrix proteins influenced the outcome (*Köller et al., 2010*).

M proteins have been reported to interact with a wide range of host proteins, such as fibrinogen (*Herwald et al., 2004*), C4b-binding protein (*Thern et al., 1995*), plasminogen (*Sanderson-Smith et al., 2006*), fibronectin (*Cue et al., 2001*), collagens (*Dinkla et al., 2003*), and immunoglobulins (*Bessen, 1994*). Importantly, binding activities vary between *emm* types. Phylogenetic clusters of M proteins broadly share binding propensities (*Sanderson-Smith et al., 2014*). However, very few M protein interactions have been confirmed through biophysical and structural analyses. The M1 protein binds to the fibrinogen D domain in the variable 'B-repeat' region (*Macheboeuf et al., 2011*). Several M protein types have the ability to bind to the C4b-binding protein through their HVRs (*Buffalo et al., 2016*). While these two molecular complexes rely on the dimeric coiled-coil topology of the M

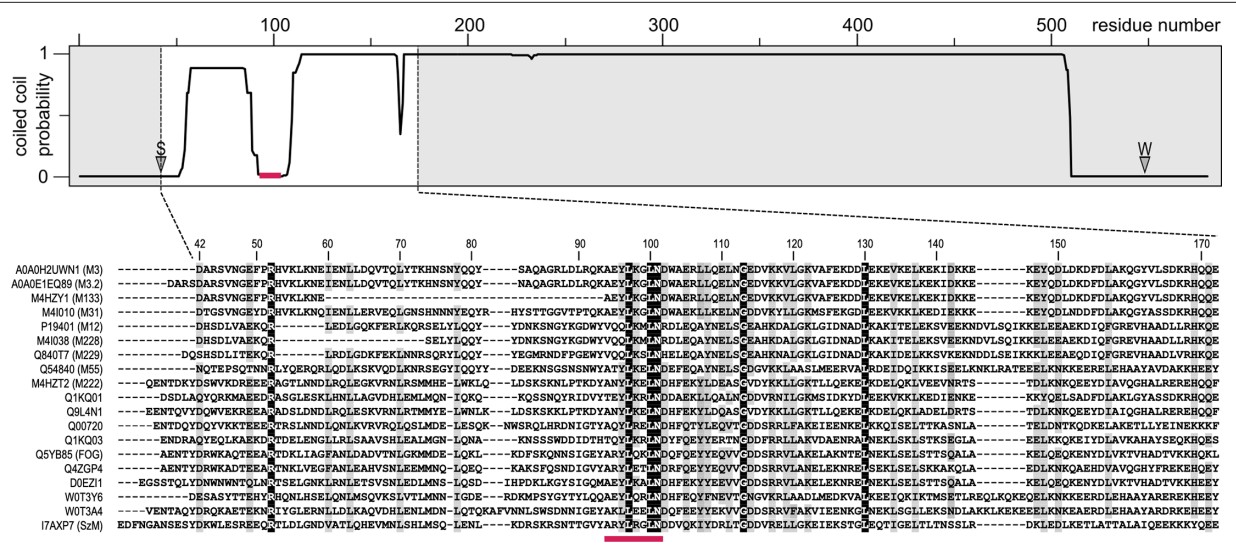

**Figure 1.** Multiple sequence alignment of the hypervariable regions of streptococcal M proteins. The top panel shows a MARCOIL coiled coil prediction (*Delorenzi and Speed, 2002*) for full-length M3 protein. Positions of signal and wall anchor cleavage sites are indicated by S and W, respectively. The PARF motif (position indicated by pink bar), previously suggested to be required for collagen binding, resides in a region of M3 that is predicted to not adopt coiled coil topology. In the alignment, sequence similarity is indicated by gray shading for residues conserved or similar in at least 75% of sequences. 100% conserved residues are highlighted by black shading. UniProt identifiers are shown to the left of the alignment. Sequences are ordered by alignment score. The proteins are from the following species: *Streptococcus pyogenes*: A0A0H2UWN1 (M3), A0A0E1EQ89 (M3.2), M4HZY1 (M133), M4I010 (M31), P19401 (M12), M4I038 (M228), Q840T7 (M229), Q54840 (M55), M4HZT2 (M222); *Streptococcus dysgalactiae* subsp. *equisimilis*: Q1KQ01, Q9L4N1, Q00720, Q1KQ03, Q5YB85, Q4ZGP4, D0EZI1, W0T3Y6, W0T3A4; *Streptococcus equi* subsp. *zooepidemicus*: I7AXP7.

proteins, the plasminogen kringle-2 domain binds to monomeric, non-coiled segments of certain M proteins (*Quek et al., 2019*; *Yuan et al., 2019*).

In a screening of GAS strains belonging to 43 *emm* types, only *emm3* and *emm18* strains were identified by Dinkla et al. to bind collagen IV to their surface (*Dinkla et al., 2003*). For *emm3* isolates, this interaction was demonstrated to depend on direct binding of collagen to the M3 protein. For *emm18* GAS strains, binding was mediated by the hyaluronic acid capsule rather than the M18 protein. Direct collagen binding has also been reported for M1 protein (*Bober et al., 2011*). Based on available evidence, collagen binding is not a universal property of GAS M proteins, but appears to be common among M proteins of group C and G streptococci (*Streptococcus dysgalactiae* subsp. *equi-similis* [SDSE]) (*Barroso et al., 2009*; *Dinkla et al., 2007*), which are emerging as important human pathogens that share disease manifestations and virulence traits with GAS (*Brandt and Spellberg, 2009*; *Xie et al., 2024*).

Collagens are commonly targeted by a wide range of pathogenic bacteria for the purposes of tissue colonization and dissemination (*Singh et al., 2012*; *Arora et al., 2021*). They have been shown to interact directly with several bacterial adhesins, such as CNA from *Staphylococcus aureus* (*Zong et al., 2005*), YadA from *Yersinia enterocolitica* (*Leo et al., 2010*), and CNE from *Streptococcus equi* subsp. *equi* (*van Wieringen et al., 2010*). In the latter two cases, adhesin binding has been mapped using the peptide libraries applied in the present study. While the biological role of collagen binding by GAS is unknown, it has been linked to the induction of an autoimmune response associated with post-streptococcal sequelae (*Dinkla et al., 2003*; *Tandon et al., 2013*). The collagen IV binding site in M3 was mapped to the N-terminal half of the protein (*Dinkla et al., 2003*). Subsequently, a collagen-binding sequence motif that is conserved in the HVRs of some M proteins of group A, C, and G streptococci was identified – the 'peptide associated with rheumatic fever' (PARF) (*Dinkla et al., 2007*). This motif is found in a region predicted to deviate from the canonical coiled coil structure in M3, several other M proteins and their homologs in non-group A streptococci (*Figure 1*). Binding of M3 protein to other collagen types has not been reported, but the M protein FOG (aka Stg11) of SDSE was found to bind to collagens I and IV (*Dinkla et al., 2007*; *Nitsche et al., 2006*).

To gain insights into mechanism and biological role of streptococcal collagen binding, we characterized the interaction of M3 with the collagen ligand collections (CLCs, triple-helical collagen peptide

libraries formerly known as Toolkits), and determined crystal structures of the M3 N-terminal domain (M3-NTD), encompassing the HVR, alone and in complex with a CLC-derived collagen peptide. We find that M3-NTD folds into a novel T-shaped domain that binds promiscuously to collagen triple helices. Furthermore, we demonstrate that the M3-collagen interaction underpins biofilm formation by GAS *emm3* isolates from necrotizing soft tissue infections, which can be inhibited in vitro by recombinant M3 protein.

## Results

### Recombinant M3 protein binds to the triple-helical domain of collagens II and III

The first aim of this study was to confirm the collagen binding activity of streptococcal M3 protein and identify any specific binding sites in collagen. To reduce the complexity of the experimental system, we chose collagens II and III as these are homotrimers, in contrast to, for instance, heterotrimeric collagens I and IV (*Ricard-Blum, 2011*). Recombinant M3 protein, lacking the N-terminal secretion signal and the C-terminal membrane-spanning region, was produced in *Escherichia coli* as a glutathione

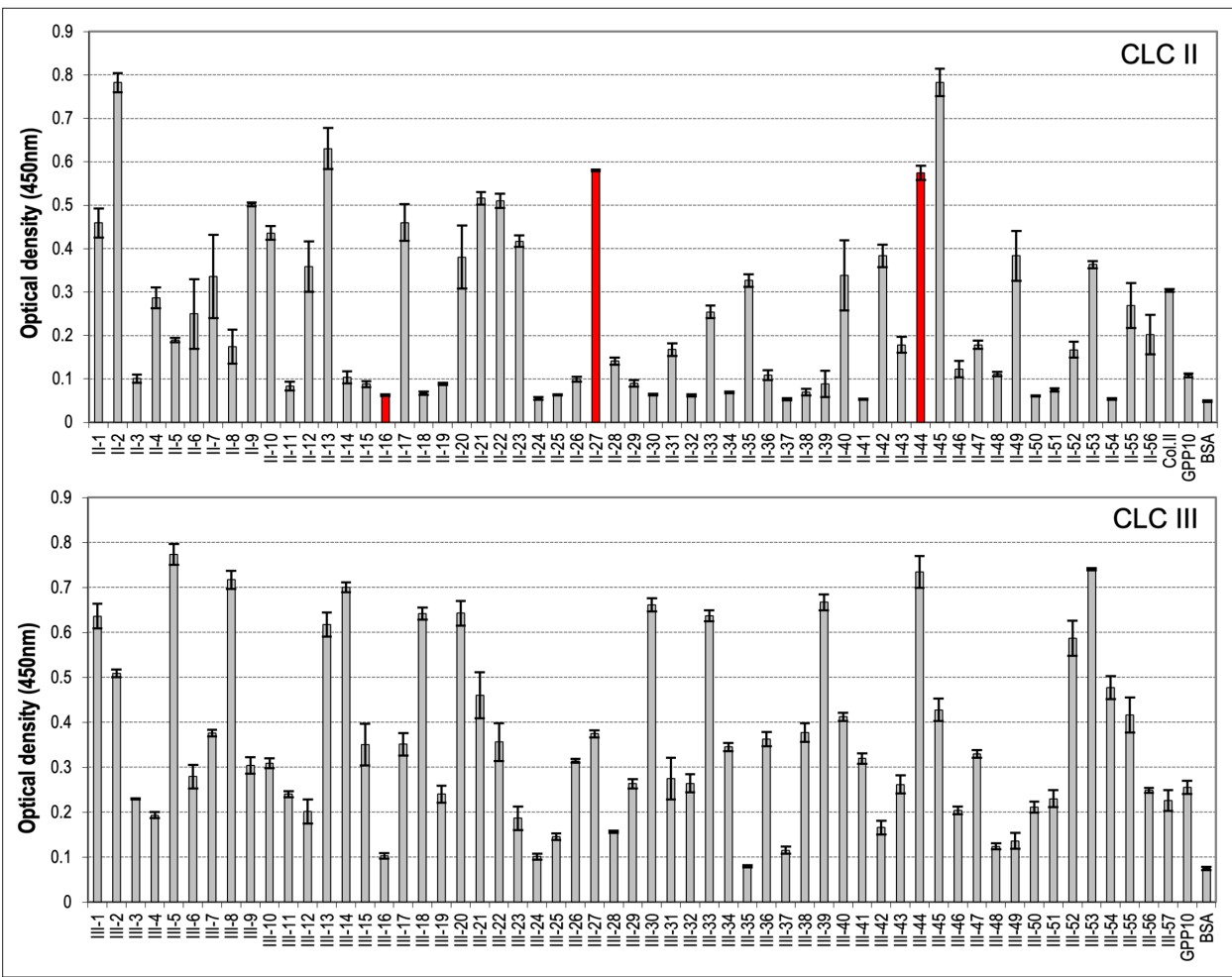

**Figure 2.** Binding of full-length M3 protein fused to GST to immobilized collagen ligand collections (CLC) II and III peptides. ELISA signal based on GST antibody is plotted against peptides spanning the tropocollagen (triple-helical) regions of collagens II and III, as well as positive (collagen II) and negative (GPP10, BSA) controls. Signals for three peptides chosen for further binding experiments are highlighted in red. Mean absorbances and standard errors of the means calculated for three replicate wells are shown.

The online version of this article includes the following figure supplement(s) for figure 2:

**Figure supplement 1.** The figure shows the number of specific residues or residue classes per collagen ligand collections (CLC) peptide (y-axis) plotted versus rank group (x-axis).

**Table 1.** Effect of amino acid classes on binding of M3 to collagen ligand collections.

| Amino acid class | Effect on binding rank | Significance |
| --- | --- | --- |
| Hydrophobic (F, L, V, M, I) | Positive | p=0.0002 |
| Hydroxyproline (O) | Positive | p=0.0003 |
| Polar (S, T) | None | |
| Polar (Q, N) | None | |
| Basic (R, K) | None | |
| GPO triplets | None | |
| Acidic (D, E) | Negative | p=0.001 |
| Proline (P) | Negative | p=0.0007 |

S-transferase (GST) fusion protein. Binding to collagens II and III was investigated using CLCs, libraries of 56 and 57 overlapping triple-helical peptides, respectively, covering the entire triple-helical tropo-collagen domain of these two collagen types (1014 and 1029 amino acids) (*Konitsiotis et al., 2008*).

Recombinant M3 was found to bind to CLC peptides to various degrees, as quantified by an ELISA-like approach using an anti-GST antibody (*Figure 2*). While there were clear differences in binding between the CLC peptides, it was impossible to identify sequence features that were conserved in good binders (high apparent affinity) but absent from peptides that gave rise to signals comparable to negative controls. Neither sequence motifs nor amino acid composition of the peptides (e.g., presence of hydroxyprolines, charged, hydrophobic, or aromatic residues) was distinct in good binders. To investigate further the sequence requirements for M3 binding, we ranked the CLC peptides in decreasing apparent affinity, measured as $A_{450}$ determined in the solid-phase binding assay, and then analyzed the distribution of particular amino acids in the entire set, pooling data from CLC-II and CLC-III. After background subtraction, we defined three binding groups: high affinity, having $A_{450}$ between 0.5 and 0.75; medium affinity, $A_{450}$ from 0.25 to 0.5; and low affinity, $A_{450}$ from 0 to 0.25.

We counted residues of each type in these three groups and compared their occurrence in peptides of the three binding groups using non-parametric tests (*Figure 2—figure supplement 1*). The outcomes are summarized in *Table 1*. Hydrophobic amino acids and hydroxyprolines appear to be more frequent in good binders, while prolines and acidic residues are underrepresented. The negative effect of proline on binding may explain why the [GPP]$_5$ flanking sequences of the CLC peptides do not dominate binding to M3, allowing marked sequence selectivity to be observed in the binding assays.

## M3-NTD harbors the collagen binding site

Full-length M proteins are not amenable to high-resolution structural characterization due to their anisotropic shape and potential conformational dynamics. We therefore designed a construct representing the M3 N-terminal domain (M3-NTD), which comprises the HVR and a short region predicted to adopt a dimeric coiled-coil structure. This fragment would not only allow structural characterization but could also be used to confirm the previously suggested localization of the collagen binding site at the HVR (*Dinkla et al., 2007*). M3-NTD includes the 110 N-terminal residues of mature M3 (residues 42–151 of the M3 protein precursor sequence). To stabilize a dimeric conformation, Leu151 was replaced with a cysteine for disulfide bond formation at the C-terminus. Leu151 is predicted to occupy a 'd' position in the canonical coiled coil heptad pattern, forming part of the hydrophobic interface between the α-helices, a position ideally suited for disulfide bond formation (*Zhou et al., 1993*). A $^{15}$N isotopically labeled version of M3-NTD was made for NMR spectroscopic analysis. From a comparison of $^{1}$H, $^{15}$N heteronuclear single-quantum coherence (HSQC) spectra, it is evident that a significant structural change occurred upon oxidation (resulting in disulfide bond formation) of the protein. The higher dispersion of signals, most noticeably in the $^{1}$H dimension, indicates disulfide-linked M3-NTD adopts a folded conformation (*Figure 3A*). In contrast, the reduced, monomeric form gives rise to a poorly resolved spectrum with signals falling within the random-coil chemical shift range, and big differences in cross-peak intensities that suggest dynamic behavior (*Figure 3B*). This demonstrates the

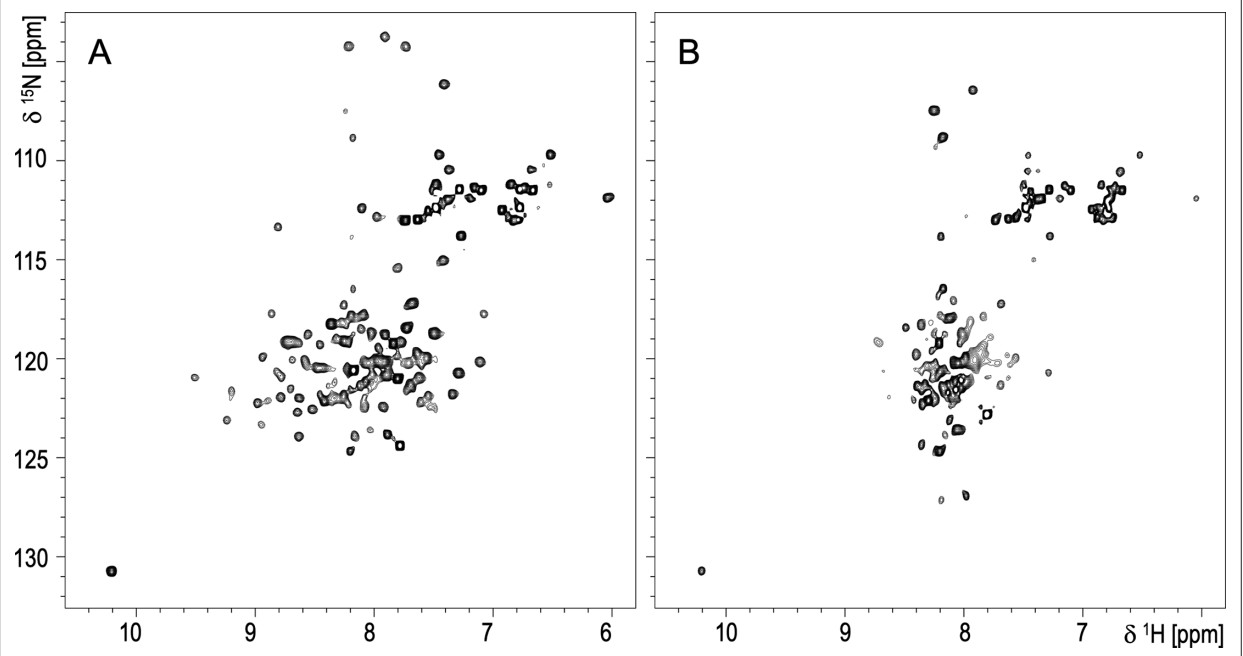

**Figure 3.** $^1$H,$^{15}$N HSQC spectra for (**A**) disulfide-bond stabilized dimeric M3-NTD and (**B**) the reduced, monomeric form.

C-terminal disulfide bond is required to stabilize a dimeric, folded structure that differs from extended unfolded or linear α-helical conformations.

To test if M3-NTD harbors the collagen binding site, binding to selected CLC triple-helical peptides was studied by isothermal titration calorimetry (ITC). We selected two peptides that showed medium to high apparent affinity in the CLC screening (II-27 and II-44) and a low-affinity peptide (II-16). These three peptides all interacted with M3-NTD in solution (*Figure 4A–C*). II-27 and II-44 binding was characterized by dissociation constants ($K_D$) in the low micromolar range (7 and 5 µM, respectively). II-16 had an approximately 10 times lower affinity ($K_D$ = 70 µM) for M3-NTD. These $K_D$s reflect the differences in binding of full-length M3 evident from the CLC solid-phase binding assay. Fitting of the sigmoidal binding curves for II-27 and II-44 suggested two indistinguishable collagen binding sites per M3 dimer. To assess if the interaction was dependent on the triple-helical conformation of the CLC peptides, we titrated M3-NTD into a monomeric version of II-44 with scrambled GPP repeat sequences at the termini (*Howes et al., 2014*). No binding was observed (*Figure 4D*). To test if collagen peptide binding required the M3-NTD to adopt its dimeric fold, the reduced (monomeric) form of M3-NTD was titrated into trimeric II-44 (*Figure 4E*). The titration curve reflected weak binding, with data not suitable for fitting. Weak binding is in line with the observation that peptides representing the PARF motif in collagen-binding M proteins have some ability to bind to collagen IV in a spot-membrane assay (*Dinkla et al., 2007*).

In conclusion, the collagen binding site of M3 was confirmed to reside in M3-NTD. CLC solid-phase binding results obtained for full-length M3 align with in-solution interaction analyses by ITC using M3-NTD. Binding of M3-NTD to collagen peptides depends on their triple-helical structure and on the dimeric conformation of M3-NTD.

## M3-NTD adopts a folded structure deviating from dimeric coiled coil

M3-NTD crystallized in several conditions with the best crystals diffracting X-rays to a resolution of 1.9 Å at the synchrotron radiation source. Phasing and structure determination was achieved using a selenium single-wavelength anomalous diffraction (Se-SAD) dataset at 2.6 Å resolution with the final model refined against the native data set at 1.9 Å resolution (*Table 2*).

M3-NTD is a symmetrical homodimer, linked at the C-terminus by a disulfide bond (*Figure 5A*). The first three residues of the construct (Gly-Ala-Met), a cloning artifact, are not resolved in the structure.

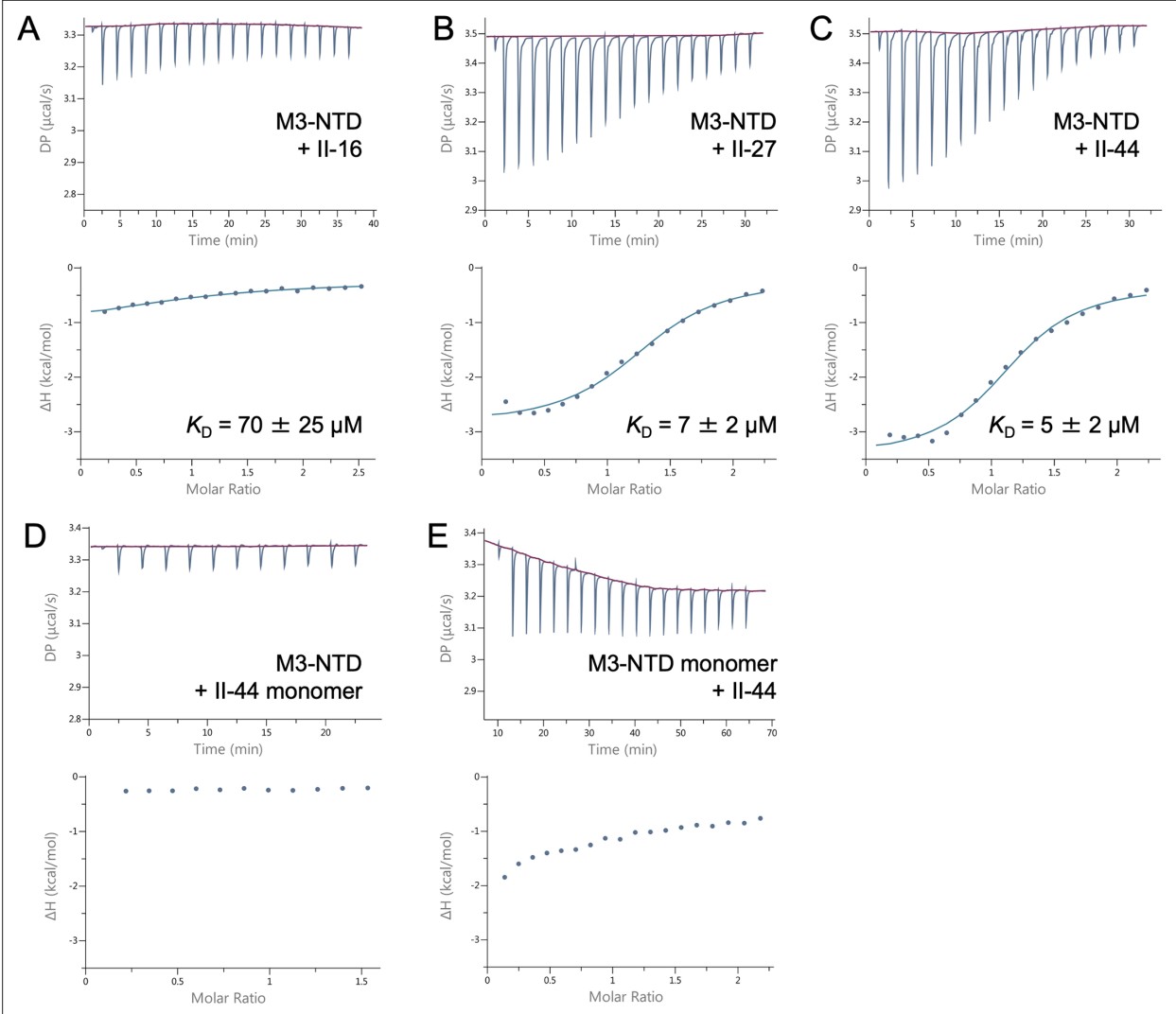

**Figure 4.** Isothermal titration calorimetry (ITC) binding curves for M3-NTD interactions with collagen ligand collections (CLC) peptides. Top panels show heat responses to repeated injections of M3-NTD into collagen peptide solutions, with baselines shown in red. Bottom panels show integrated signals (enthalpy changes) plotted against the molar ratio of binding partners. Where nonlinear fitting gave meaningful results, the fits are shown as blue lines, and dissociation constants ($K_D$) are specified. (**A**) ITC data for interaction of M3-NTD with triple-helical peptide II-16. (**B**) ITC data for interaction of M3-NTD with triple-helical peptide II-27. (**C**) ITC data for interaction of M3-NTD with triple-helical peptide II-44. (**D**) ITC data for interaction of M3-NTD with monomeric peptide II-44. (**E**) ITC data for interaction of monomeric M3-NTD with triple-helical peptide II-44.

The fold represents a novel T-shaped architecture with no significantly similar structures identifiable by the protein structure comparison server DALI (*Holm et al., 2023*). Each monomer is composed of three α-helices, H1-H3. The H3 helices comprise the C-terminal 38 residues of M3-NTD (M3 residues 114–151) and form a coiled coil stem. This, in the full-length protein, would be extended into an ~50-nm-long coiled coil characteristic of M proteins (*Figure 5B*). Helices H1 and H2 form hairpins that pack against each other to form the slightly kinked bar of the T-shape, effectively a three-helix bundle. A key residue is Gly113, which resides at the T-junction, separating H2 and H3 (*Figure 5C*). It is completely conserved in M proteins identified in our sequence similarity search (*Figure 1*). The T-junction structure is defined by conserved leucine and isoleucine residues that form a small hydrophobic core (*Figure 5C*). The T-bar structure is stabilized by a network of polar interactions of several conserved residues, most notably inter-chain salt bridges of Arg52 with Asp102 and Glu105, and an inter-chain hydrogen bond between the side chains of Glu49 and Asn101 (*Figure 5D*). Asn101 is one of the completely conserved residues in the PARF motif, previously implicated in collagen binding. This motif is located on H2 at the bottom of the T-bar (*Figure 5A*). Some but not all conserved PARF

**Table 2.** Data collection and refinement statistics.
Values in brackets refer to the highest resolution shell.

| | M3-NTD | SeMet M3-NTD | M3-NTD+JDM238 |
|---|---|---|---|
| Data collection | 8P6K | | 8P6J |
| Spacegroup | C2 | C2 | P1 |
| Cell dimensions | | | |
| a, b, c (Å) | 150.5, 24.1, 78.0 | 150.4, 25.1, 77.0 | 31.8, 51.5, 80.2 |
| a, b, c (°) | 90, 105.3, 90 | 90, 104.4, 90 | 87.3, 84.8, 89.4 |
| Resolution (Å) | 37.62–1.92 (1.94–1.92) | 37.3–2.67 (2.71–2.67) | 79.77–2.32 (2.45–2.32) |
| $R_{meas}$ | 0.058 | 0.198 | 0.107 |
| I/σ | 14.8 | 10.2 | 2.1 |
| $CC_{1/2}$ | 0.9 (0.5) | 1 (0.9) | 1 (0.3) |
| Completeness (%) | 97.80 (94.07) | 98.8 | 89.58 (89.66) |
| Multiplicity | 6.2 | 18.6 | 1.7 |
| Refinement statistics | | | |
| Resolution (Å) | 37.6–1.92 | | 79.77–2.32 |
| No. reflections | 21411 (1054) | | 19530 (947) |
| $R_{work}$/$R_{free}$ | 0.230/0.268 | | 0.241/0.285 |
| Ramachandran favored (%) | 97.7 | | 97.7 |
| Ramachandran outliers (%) | 0 | | 0.5 |
| RMS deviations | | | |
| Bond lengths (Å) | 0.039 | | 0.013 |
| Bond angles (°) | 2.06 | | 1.89 |

residues contribute to the formation of the T-bar structure of M3-NTD. Leu97 and Asn101 are the only completely conserved residues of PARF with a structural role. Leu97 is buried at the interface between the H1-H2 hairpin of one monomer and H1 of the other monomer, packing against the Tyr81 side chain (*Figure 5D*).

## An N-terminal T-shaped fold is predicted to be a feature of other M proteins

AlphaFold3 (*Abramson et al., 2024*) predictions were carried out for the proteins identified in our sequence similarity search (*Figure 1*) to support the structural role of the conserved residues. In validation of this approach, the AlphaFold3 predicted model of M3 is almost perfectly superimposable with the experimental M3-NTD structure (*Figure 6A*). All other proteins included in this study are predicted to have T-shaped N-terminal domains, with the exception of proteins M133 and M228, where deletions break the topology of the T-bar fold (*Figure 6B*, *Figure 6—figure supplement 1*). Some of the predictions for GAS M protein variants (M12, M55, M229) carry low confidence, and these proteins may well not adopt the T-fold. HVRs of M proteins that are phylogenetically more distant to M3, such as M1 and M28, are not predicted to fold into any distinct tertiary structure deviating from coiled coil (*Figure 6C*). On the other hand, M proteins of SDSE and the SzM protein of *Streptococcus equi* subsp. *zooepidemicus* (SESZ) are predicted with high confidence to adopt a structure similar to M3. This analysis structurally validates our sequence alignment of M3 homologs. It suggests the T-shaped structure of the NTD is a feature of a subclass of M proteins and is common in SDSE.

## M3-NTD binds promiscuously to the collagen triple helix

M3-NTD is a structurally tractable fragment that retains the collagen binding activity of the full-length protein. We chose II-27, a CLC peptide with good affinity for M3-NTD, as the basis for structural

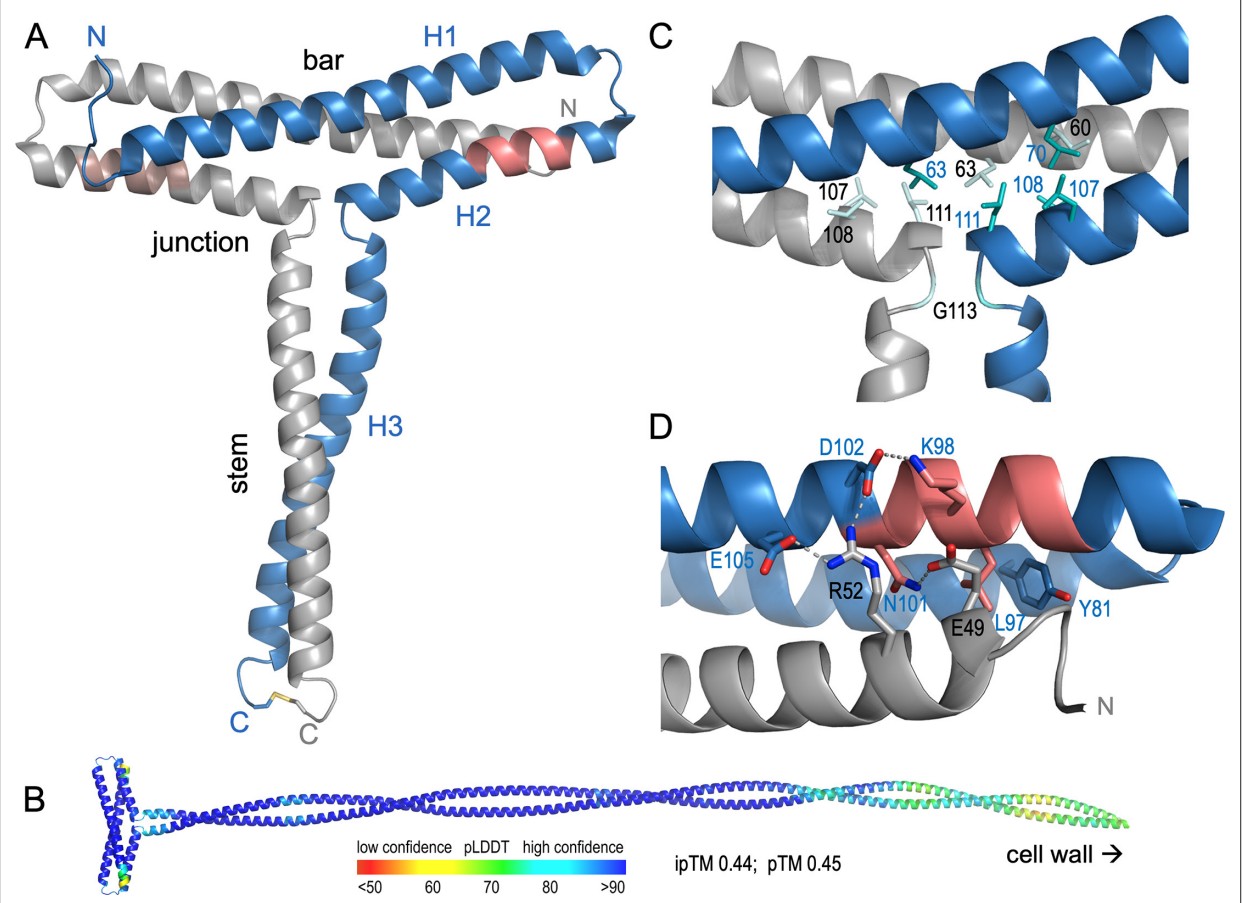

**Figure 5.** Crystal structure of M3-NTD (PDB 8p6k). (**A**) Ribbon diagram of the covalently stabilized M3-NTD dimer. Monomers are shown in blue and gray. The regions previously associated with collagen binding (PARF motif) are highlighted in red. N-termini (Asp42) and C-termini (Cys151) are labeled N and C, respectively. The three helices of the blue monomer are labeled H1-3. The C-terminal disulfide bond is shown as sticks. (**B**) Ribbon diagram model for the full extracellular region of M3, predicted by AlphaFold3 (*Abramson et al., 2024*) and colored by per-residue confidence (pLDDT). (**C**) Conserved leucine, isoleucine (Ile60), and glycine (Gly103) residues define the T-junction structure of M3-NTD. (**D**) Role of conserved residues around the PARF motif in stabilizing the T-bar region of M3-NTD. Polar contacts are shown as dashed lines.

characterization of an M3-collagen complex. II-27 contains an atypical α1β1 integrin-selective motif, GVOGEA (*Hamaia et al., 2012*). To increase chances of crystallization, a shorter 24-residue peptide, JDM238, was synthesized which contains the GVOGEA motif flanked by three glycine-proline-4-hydroxyproline (GPO) repeats at each terminus. Crystals formed in conditions containing JDM238 and M3-NTD at a 3:2 molar ratio and diffracted synchrotron X-rays to 2.3 Å resolution. Molecular replacement with M3-NTD yielded a high-quality structural model for the complex (*Table 2*).

In agreement with our ITC data for CLC II-27 binding to M3-NTD, the M3-NTD dimer is bound to two copies of triple-helical JDM238. While both copies occupy equivalent binding sites on opposite faces of the M3-NTD T-bar, they contact the protein with two different regions (*Figure 7*). One copy contacts M3-NTD through residues 15–21 of the C-terminal GPO repeats, the other is bound at its more central GPOGVO region (residues 6–12) (*Figures 7 and 8A*). The presence of two different binding registers in the same crystal is likely an effect of crystal packing, as this binding mode results in a compact assembly required for crystallization (*Figure 7—figure supplement 1*), but it also reflects a lack of specificity of collagen binding by M3, in line with the CLC screening data. Superposition of the two binding sites highlights how the M3-NTD T-bar complements the characteristic and uniform surface topology of triple helices in both binding registers in the same fashion (*Figure 8B*).

The M3 collagen binding site includes the PARF region on H2 (residues 94–101), extending beyond it to include the C-terminal half of H1 and most of H2 (*Figure 8*). This confirms previous reports on the involvement of PARF in collagen binding (*Barroso et al., 2009*; *Dinkla et al., 2007*). Interfaces

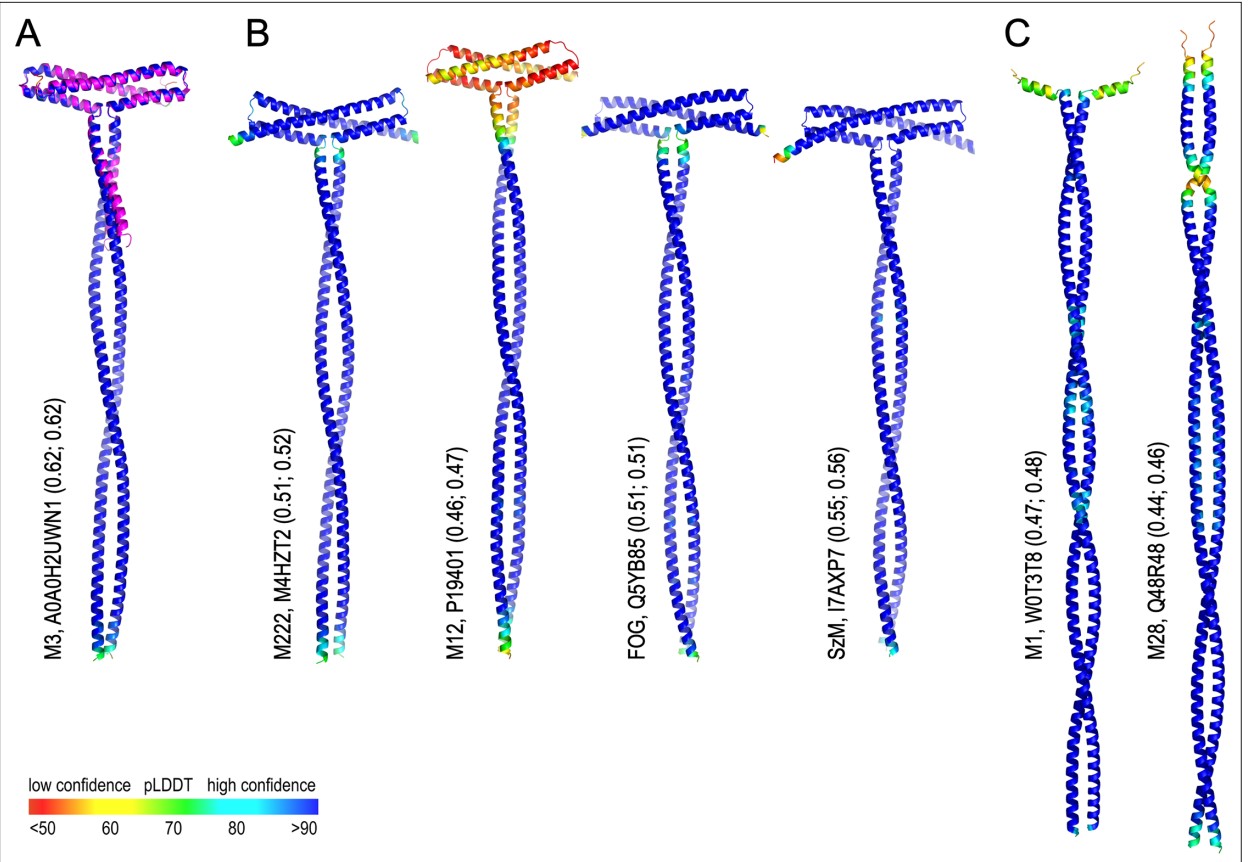

**Figure 6.** AlphaFold3 predictions for M proteins of group A streptococcus (GAS), *Streptococcus dysgalactiae* subsp. *equisimilis* (SDSE), and *Streptococcus equi* subsp. *zooepidemicus* (SESZ). (**A**) Overlay of the experimental M3-NTD structure (magenta) with a predicted structure for the N-terminal 230 residues of mature M3 (colored by pLDDT). (**B**) Predicted structures for the N-terminal 230 residues of other M proteins included in the sequence alignment (*Figure 1*). (**C**) Predicted structures for the N-terminal 230 residues of M1 and M28 proteins, which are not known to interact with collagens. Values in parentheses are AlphaFold3 prediction quality parameters ipTM and pTM, respectively.

The online version of this article includes the following figure supplement(s) for figure 6:

**Figure supplement 1.** Additional AlphaFold3 predicted structures of proteins included in the sequence alignment (*Figure 1*).

between the binding partners largely comprise hydrophobic interactions between highly complementary surfaces. M3 residues Tyr96 and Trp103 play prominent roles: they pack against Hyp and Pro sidechains, filling grooves on the collagen triple helices and form polar contacts with the peptide backbone. The indole amino group of Trp103 interacts with the carbonyl of Ala15 or Hyp6 (depending on the register), while Tyr96 forms a water-mediated hydrogen bond with the carbonyl of Hyp9 or Hyp18. In one binding mode, M3 Gln42 hydrogen bonds with the same water molecule as Tyr96. Direct polar contacts also include M3 residues Lys35 and Arg49 that interact with the hydroxyl group of Hyp9 and the backbone carbonyl of Hyp12, respectively. In addition, in one binding mode, M3 residue Gln46 forms a hydrogen bond with Gly22 of the peptide via a water molecule.

## Two M3 residues are essential for collagen peptide binding

Tyr96 is a conserved residue of the PARF motif. We tested its role in collagen binding using two site-specific variants, Tyr96Ala and Tyr96Phe. When titrating M3-NTD Tyr96Ala into CLC peptide II-44, which bound to wild type M3-NTD with low micromolar $K_D$, no binding was observed (*Figure 8—figure supplement 2*). The more conservative substitution of Tyr96 with Phe also greatly reduced the affinity for II-44 ($K_D$ ~200 μM) (*Figure 4F*), suggesting that the water-mediated interaction between Tyr96 and the peptide backbone strongly contributes to binding.

Similar results were obtained for Trp103, the second prominent residue in the M3/collagen interface. Mutation to an Ala led to a loss of binding as detected by ITC. Replacement of Trp by the large

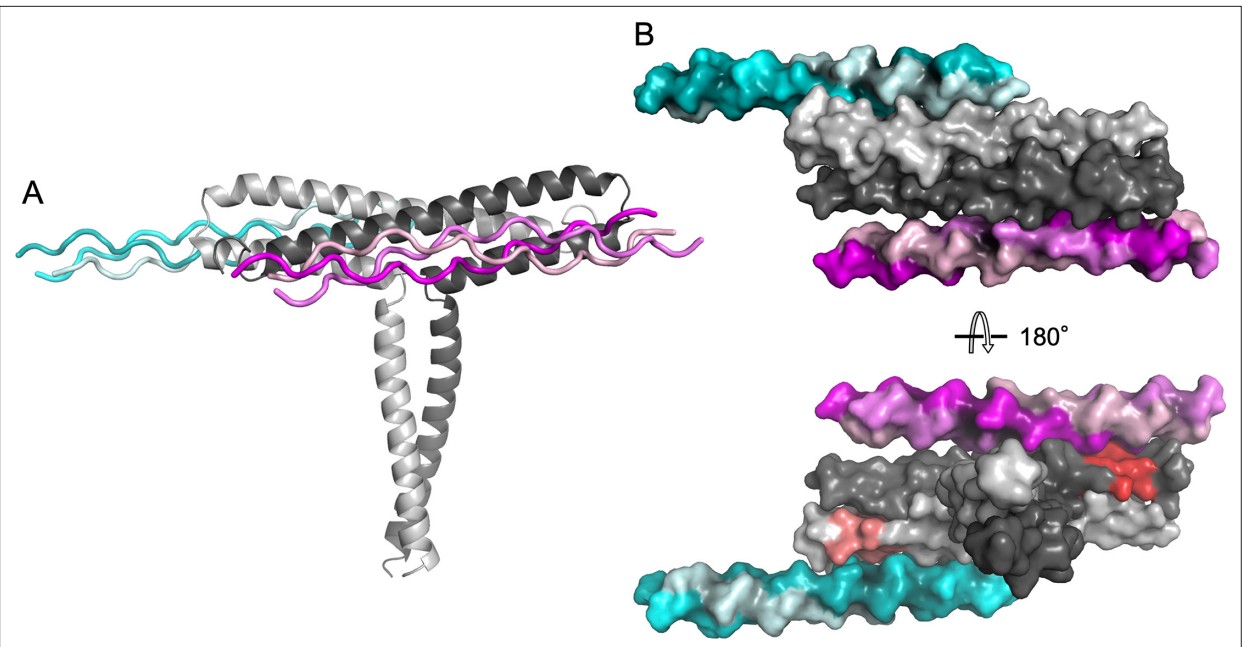

**Figure 7.** Structural basis of collagen binding by M3 (PDB 8p6j). (**A**) Crystal structure of the complex of M3-NTD (subunits shown in light and dark gray cartoon representation) with collagen-derived peptide JDM238. Two binding registers are observed bound to equivalent sites of the M3-NTD dimer, indicated by shades of magenta and cyan. (**B**) Space-filling models of views from the top of the T-bar of M3-NTD (top) and looking down the stem toward the bottom of the T-bar (bottom). The PARF motif is shown in tones of red, otherwise, coloring is as in (**A**).

The online version of this article includes the following figure supplement(s) for figure 7:

**Figure supplement 1.** Crystal packing.

hydrophobic Ile resulted in detectable but weak binding, with data not suitable for non-linear fitting to derive meaningful thermodynamic parameters (*Figure 8—figure supplement 2*).

To rule out any effect of mutations of Tyr96 and Trp103 on the structure of M3-NTD, [1]H NMR spectra were recorded (*Figure 8—figure supplement 3*). The spectra all showed very similar dispersion of resonances, reflecting a folded state with some hydrophobic core giving rise to resonances below 0.5 ppm. The only readily assignable resonance, due to its unique chemical shift, is the indole NH of Trp103, which, as expected, is lacking in the Trp mutants but found at almost exactly the same chemical shift in M3-NTD and the Tyr96 mutants. These data support the loss of affinity of collagen peptides for the Tyr96 and Trp103 mutants being attributed to contribution of these residues to the binding interface rather than a loss of structure, as would be expected from the crystal structures.

This supports the critical role of Tyr96 and Trp103 in collagen binding, both via forming van der Waals interactions with the collagen peptide triple helix as well as forming direct and water-mediated polar interactions with the peptide backbone.

In conclusion, we present the first structural evidence for an M-protein collagen complex. The structure indicates a general binding mechanism that might explain the promiscuity of M3 for diverse CLC peptides and explains how M3 is able to bind non-selectively to different collagen types, which all share very similar triple-helical (COL) domains. This is supported by an AlphaFold3 prediction of a complex between M3 and a type I collagen peptide, which is predicted to bind to M3 in a way that is highly similar to the experimental complex structure (*Figure 8—figure supplement 1B and C*).

## *emm* type-dependent effect of human collagen on biofilm formation by GAS strains

GAS *emm*3 strains are highly prevalent in necrotizing soft tissue infections (*Luca-Harari et al., 2009*; *Bruun et al., 2021*), where they have been found to form biofilm (*Siemens et al., 2016*). Based on our previous data demonstrating that the tissue milieu seems to promote GAS biofilm (*Siemens et al., 2016*), and with collagen being a ubiquitous structural protein, it was of interest to assess whether the M3-collagen interaction could affect biofilm formation. For this purpose, we used crystal violet-based

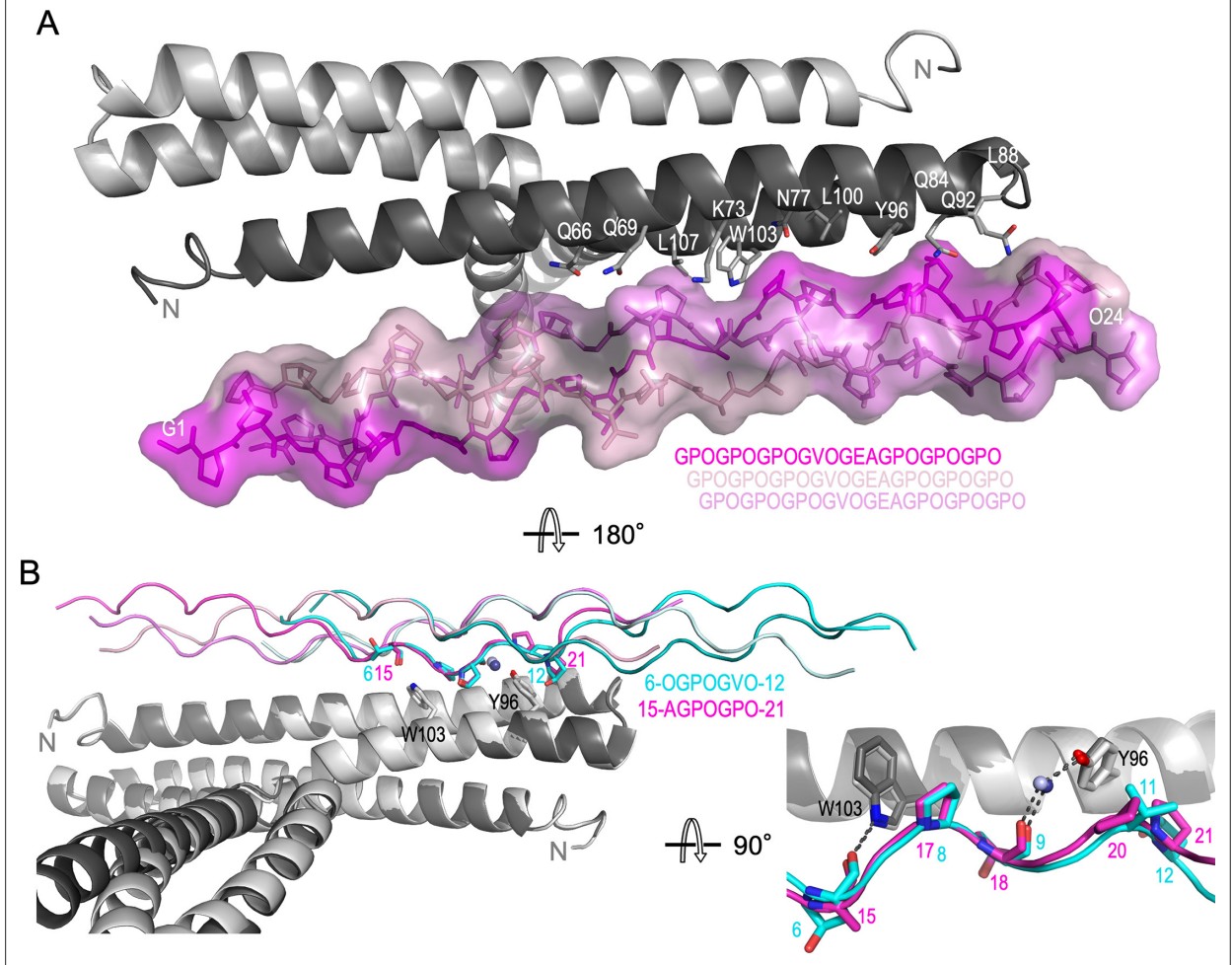

**Figure 8.** Collagen triple helix/M3-NTD interface. (**A**) Residues of M3-NTD (gray cartoon representation) that form the collagen binding site are shown as sticks and are labeled. The collagen peptide triple helix is shown in stick and surface representation in shades of magenta. The sequence is shown indicating the staggered arrangement of the chains in the triple helix. (**B**) Superposition of the two peptide binding sites to highlight conservation of collagen binding mode despite sequence deviation between the two binding registers. Tyr96 and Trp103 are shown as sticks. The two monomers of M3-NTD are shown in light and dark shades of gray. Collagen peptides of the two binding modes are shown in magenta and cyan, with five equivalent side chains shown as sticks. The equivalent sequences of the binding sites are shown. Water molecules in the binding interface are shown as blue spheres. In the zoomed-in image on the bottom right, only one of the collagen peptide chains is shown per complex for clarity. Hydrogen bonds are shown as gray dashed lines.

The online version of this article includes the following figure supplement(s) for figure 8:

**Figure supplement 1.** AlphaFold3 predicted structures of M proteins in complex with 24-residue collagen peptides.

**Figure supplement 2.** ITC binding curves for M3-NTD variants with peptide II-44.

**Figure supplement 3.** [1]H NMR spectra of M3-NTD and variants with key collagen-binding residues mutated.

assay to determine biofilm formation by GAS strains from patients with necrotizing soft tissue infections on polystyrene plates, either uncoated or coated with human type I collagen. The three isolates, *emm*1, *emm*3, and *emm*28 strains, all formed biofilm on uncoated plates (*Figure 9A*). Notably, on collagen-coated plates, biofilm of isolates 2006 (*emm*1) and 5004 (*emm*28) was significantly reduced in a dose-dependent manner, while that of isolate 2028 (*emm*3) was significantly enhanced (*Figure 9B and C*). The collagen-enhancing effect of biofilm for the *emm*3 strain was seen even at inocula as low as 100 CFU/well (*Figure 9D*), where also a dose–response to collagen was evident (*Figure 9D*). Confocal microscopy was used to assess bacterial attachment and biofilm formation with bacteria grown on glass slides with or without collagen coating. In line with the crystal-violet assay, bacterial attachment of the *emm*1 (2006) and the *emm*28 (5004) strain was almost completely blocked by

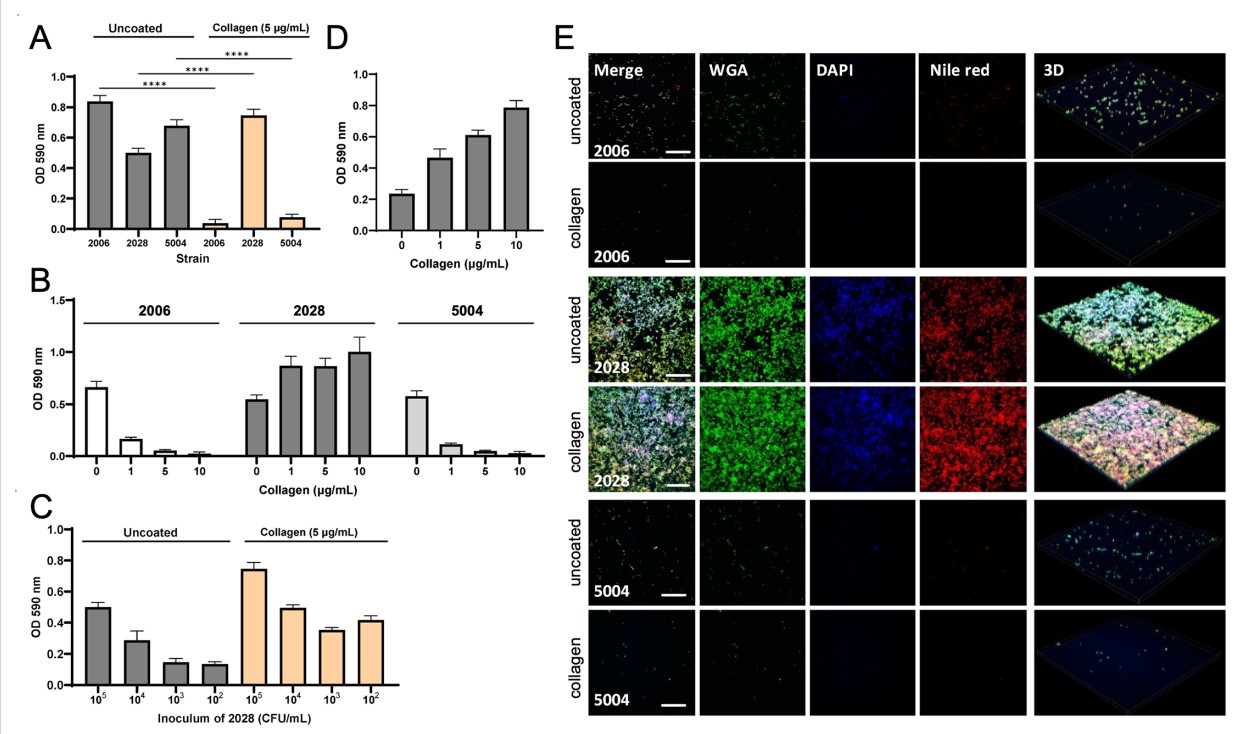

**Figure 9.** Effect of collagen type I on biofilm formation by necrotizing soft tissue infection strains. (**A**) Quantitative analysis of biofilm formation on polystyrene surface with or without human type I collagen. The inoculum of each bacterial strain was $10^5$ CFU per well. Significance was determined by one-way ANOVA with Tukey's *post hoc* test. ****$p<0.0001$; ***$p<0.001$; **$p<0.01$; *$p<0.05$ (**B**) Varying effect of collagen concentration. The inoculum of each bacterial strain was $10^5$ CFU per well. (**C**) Inoculum effects on biofilm of strain 2028. (**D**) Effect of concentration of coating collagen on strain 2028 biofilm (100 CFU per well). (**A–D**) Results are shown as mean + SE. All assays were repeated at least three times in triplicate. (**E**) Confocal microscopic analysis of biofilm formation 48 h after incubation. Fluorescence staining (WGA-Alexa 488, DAPI, and Nile red) of biofilm on uncoated and collagen-coated glass slides. Scale bars indicate 50 µm.

collagen, whereas the *emm*3 (2028) strain formed strong, robust biofilm in the presence of collagen (**Figure 9E**).

Taken together, the data show a collagen-enhanced biofilm formation for the *emm*3, but not the *emm*1 and *emm*28 strains. To test whether this effect was linked to the M3-protein/collagen interaction, we performed competition experiments where biofilm formation of the *emm*3 (2028) strain was assessed in the presence of increasing concentrations of M3-NTD in the medium. The enhancing effect of collagen was almost completely negated in the presence of 20 µM M3-NTD (**Figure 10**). These data indicate that the interaction between M3 protein and collagen enhances biofilm formation. In further support of this finding, two additional *emm*3 strains, GAS 5626 and 8003 showed a similar increase in biofilm formation on collagen-coated plates (**Figure 10—figure supplement 1A**). The expression of M3 protein by all three *emm*3 strains was confirmed using an M3-specific antibody (**Figure 10—figure supplement 1B**).

## GAS M3-collagen interaction in patient biopsies and in a 3D skin tissue model

To demonstrate a potential interaction between GAS and collagen during infection, we stained tissue biopsies from patients with necrotizing soft tissue infections using specific anti-GAS and anti-collagen IV antibodies (**Figure 11**). In areas with the presence of bacteria, collagen was observed to co-localize with the bacteria in biopsies from two patients infected with *emm*3 strains 2028 and 5020, and in one out of two patients infected with *emm*1 strains (2006 and 2068).

To further investigate this in a more controlled infection model, we used a human 3D organotypic skin model, which is based on a collagen type IV scaffold. We have previously employed this tissue model for studies of biofilm and GAS-elicited tissue pathology (***Bergsten et al., 2021***; ***Siemens et al.,***

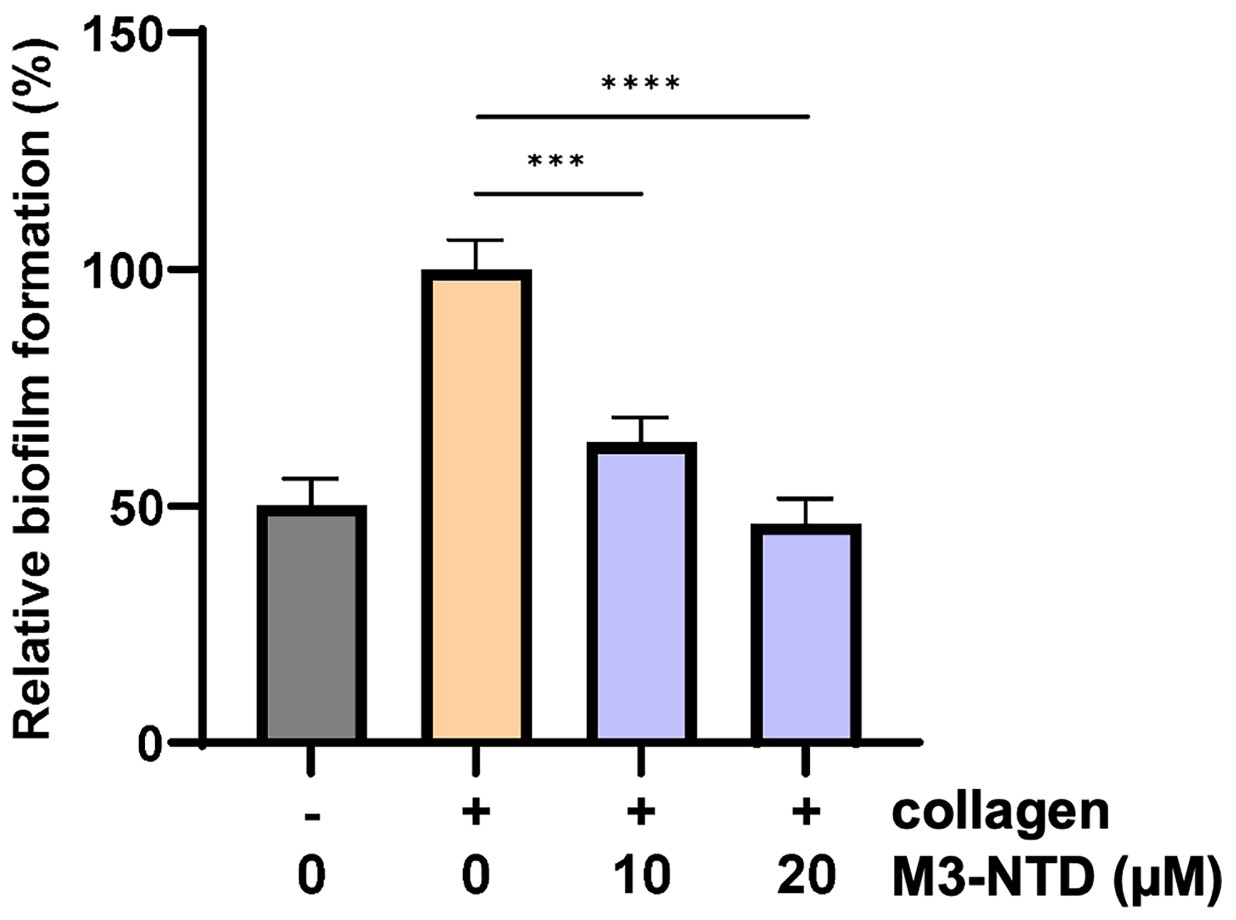

**Figure 10.** Competition analysis of biofilm formation of group A streptococcus (GAS) isolate 2028 (100 CFU per well) in the presence of M3-NTD. Results are shown as mean + SE. All assays were repeated three times in triplicate. Significance was determined by one-way ANOVA, followed by Tukey's *post hoc* test ****p<0.0001; ***p<0.001.

The online version of this article includes the following figure supplement(s) for figure 10:

**Figure supplement 1.** Biofilm formation of *emm*3 group A streptococcus (GAS).

*2015*). Infection of the skin organotypic tissue with the *emm*1 (2006) and *emm*3 (2028) strains showed that both strains efficiently infected the skin model tissue and caused disruption of the epithelial layer (*Figure 12*). Bacterial load and tissue pathology increased over time, as did the colocalization of *emm*3 with collagen. However, no such increase was noted in the *emm*1 strain-infected models. These data suggest that an interaction between *emm*3 GAS and collagen occurs in human tissue and might be an important factor for biofilm formation.

## Discussion

No obvious or simple connection can be made between M protein sequences and the differential abilities of the over 200 variants to interact with host factors. At least some activities of this major streptococcal virulence factor may be encoded in hidden sequence patterns, as identified for the C4b-binding protein interaction with HVRs (*Buffalo et al., 2016*). In this study, we show that the HVR of the M3 protein deviates from the canonical coiled-coil structure generally assumed to be adopted by M proteins. The novel T-shaped fold identified here is required for the collagen binding activity of M3. While this fold may be limited to a few phylogenetically close variants of M proteins in GAS, it is more abundant in SDSE. The discovery of a folded HVR may inform vaccine design. Although immunization with short and therefore monomeric M protein HVR peptides may be sufficient for most GAS sero-types, immunogenic epitopes of M3 may only be present in the folded, dimeric form of the protein.

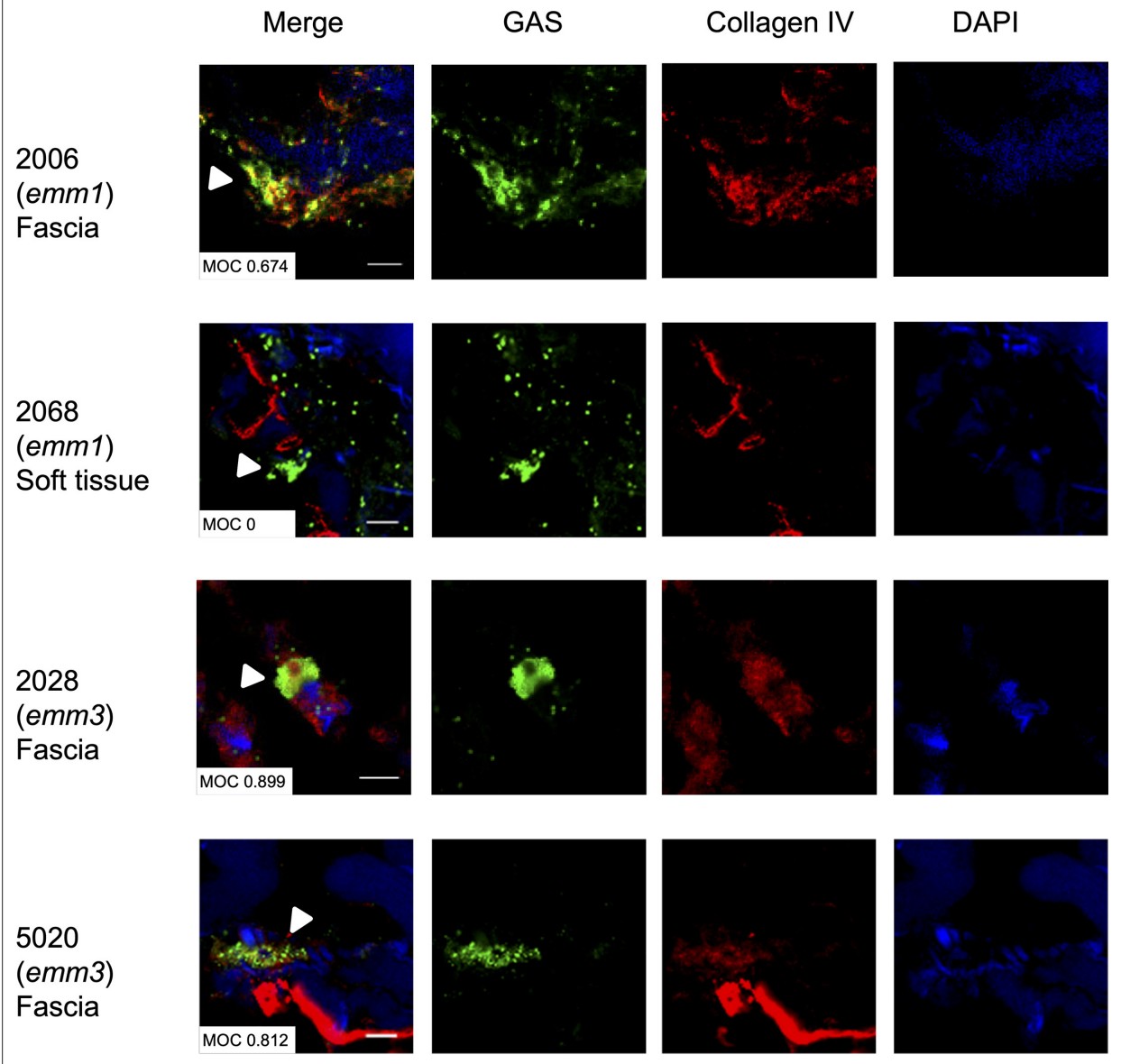

**Figure 11.** Colocalization of collagen with *emm3* group A streptococcus (GAS) in patients' biopsies. Frozen biopsy sections were stained with anti-GAS (green), anti-collagen IV (red), and DAPI (blue). Scale bars indicate 50 µm. The Mander's overlap coefficient (MOC) was used to quantify the colocalization of bacteria with collagen.

The structures of M3-NTD alone and in the presence of a collagen peptide give a rational basis for the role of the PARF motif, which was previously implicated in collagen binding (*Dinkla et al., 2007*). The role of conserved residues within PARF is predominantly to stabilize the T-fold of M3-NTD. The motif alone, encompassing eight residues (Ala94 to Asn101) as defined by *Barroso et al., 2009*, does not present all structural features involved in collagen binding based on our structure. This depends on a larger region spanning residues Gln92 to Leu107 on the H2 helix, with some minor contributions of residues in H1 (*Figure 8*). The complex structure identifies Tyr96 and Trp103 as key interface residues. They provide shape complementarity and stack against proline and hydroxyproline rings of the collagen triple helix. Our mutational analysis confirms a role in collagen binding for Tyr96 and Trp103. Conclusions from the crystal structure are consistent with our analysis of the binding propensities of the different amino acids in the CLC peptides. The observed prominent roles for hydrophobic residues and for hydroxyproline in the solid-phase binding assays would be predicted from the complex structure. The negative effects on binding rank of glutamate and of proline are less easy to explain.

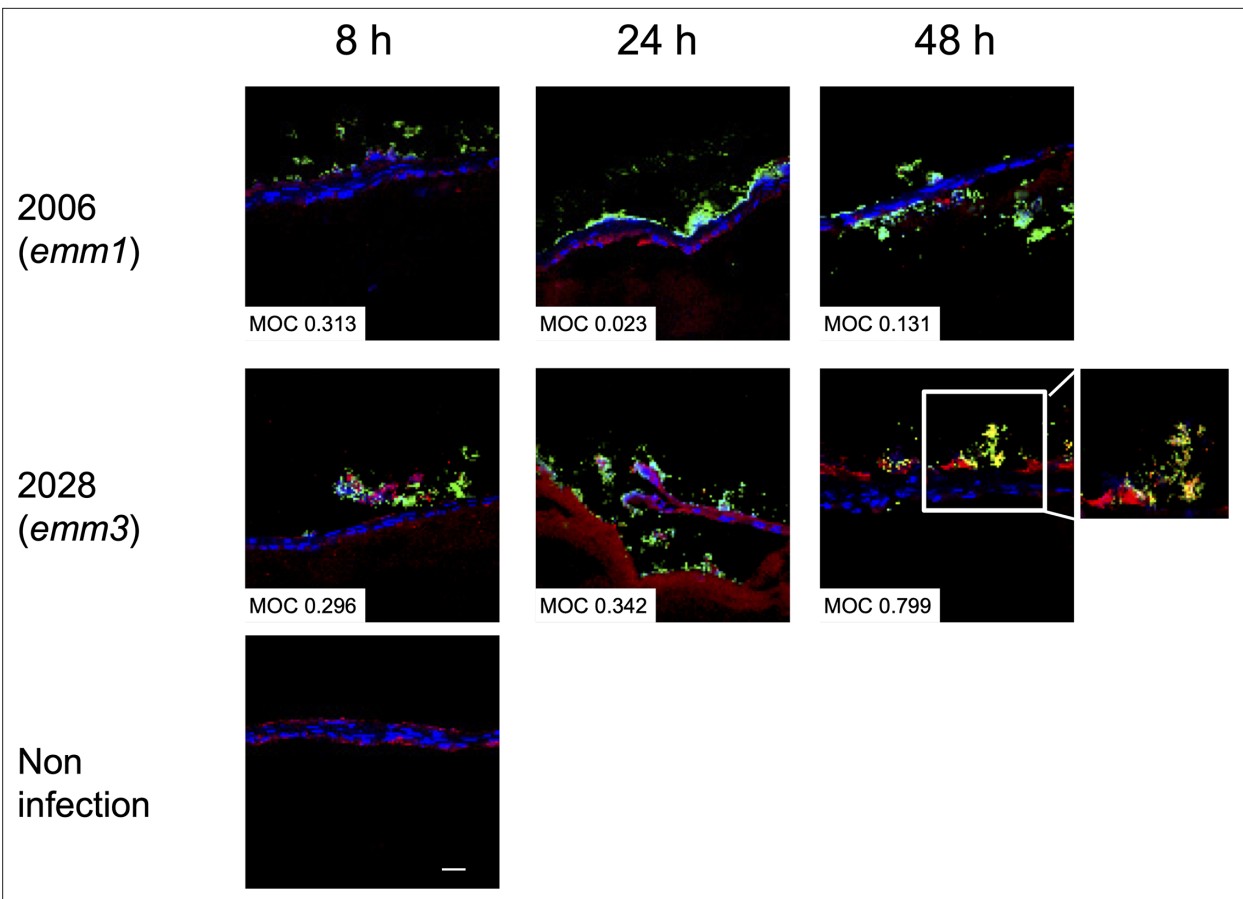

**Figure 12.** Colocalization of collagen with *emm3* group A streptococcus (GAS) in 3D skin tissue model. Skin tissue models were infected with GAS strains 2006 and 2028. At 8, 24, and 48 h after infection, frozen sections were stained with anti-GAS (green), anti-collagen IV (red), and DAPI (blue). The Mander's overlap coefficient (MOC) was used to quantify the colocalization of bacteria with collagen. The merged images are shown.

Proline is restricted to the X position of the GXY triplet, which is also the position usually occupied by glutamate. The effect of both residues may be to exclude more productive hydrophobic amino acids from the X position. The positive role of hydroxyproline in the Y position is amply demonstrated in the crystal structure. In addition to its ability to form hydrogen bonds, this may derive from its greater extension from the helix axis than proline. Our findings are reminiscent of those obtained for YadA (*Leo et al., 2010*), in that we also observed promiscuous binding to the CLC peptides, which was strongly dependent on hydrophobic residues. However, in the case of YadA, the number of GPO triplets and number of both Pro and Hyp also correlated with affinity.

The lack of specificity of CLC peptide binding and the alternative binding registers observed in the crystal structure of the M3-NTD complex provide a rationale for the ability of M3 and related proteins to target various types of collagens. The M3 fold recognizes the collagen triple helix through shape complementarity. While collagen types differ greatly in their structural organization and biological functions, their defining feature is the presence of triple-helical COL domains. The most abundant collagens are the fibrillar collagens I, II, III, and V, which share high sequence homology and are rich in the hydrophobic and hydroxyproline residues needed to bind M3. The non-fibrillar collagens IV and VI have a similar amino acid content. All these abundant collagen types contain extensive triple-helical domains that we show can bind M3 and, crucially, are found in the epithelial and endothelial surfaces where they may encounter GAS.

The molecular complex described here explains how GAS of the M3 type achieve very strong binding to collagens. While the affinity for the triple-helical peptides measured here by ITC is moderate, the presence of two binding sites on the HVR that is found at the distal end of each M3 protein suggests cooperative binding in the context of larger collagen assemblies. Most likely,

the bacteria do not encounter isolated triple helices, but higher-order collagen structures made of bundles of triple helices. It can be envisaged how M3 proteins probably intercalate between triple helices in such higher-order assemblies. Given the high density of M proteins on GAS surfaces, this would generate a polyvalent, high-avidity host-bacteria interface with the HVR acting as an anchor binding M3 perpendicularly to collagen fibrils/fibers. It is also at this higher-order structural level that an explanation for the ability of M3 to induce an anti-collagen autoimmune response might be found. In the rare autoimmune disease Goodpasture syndrome, an anti-collagen response is thought to be caused by collagen IV neoepitopes, with infections being suggested as one possible cause for their exposure (*Pedchenko et al., 2018*). It is conceivable that such neoepitopes could be generated as a result of collagen structural changes caused by M3 protein binding. However, such structural rearrangements could not be observed in our simplified molecular system, and our data cannot provide evidence backing the anti-collagen hypothesis of post-streptococcal rheumatic sequelae (*Tandon et al., 2013*) beyond providing the molecular basis for collagen recognition by M3 protein.

The structural data presented here allow us to conclude that collagen binding by M proteins relies on the presence of a T-shaped helical bundle domain combined with key residues to provide shape complementarity to tropocollagen triple helices. But even if all GAS M protein variants included in our sequence analysis bound collagens in a similar manner to M3, this would still indicate that M3-like collagen binding mechanism is rare among GAS M protein variants. Based on data published for M55, which was found to bind comparably weakly to collagen IV (*Reissmann et al., 2012*), it seems likely that M3 and closely related serotypes have a unique ability to bind collagens in the way described here. AlphaFold3 predictions support this, yielding converging structural models for the M3-collagen peptide complex that closely resembled our experimental structure. This was also true for M31 and M222. However, predictions with M12 and M55 resulted in low-confidence, non-converging structural models with poorly defined binding sites, and/or disruption of the T-fold (*Figure 6—figure supplement 1*). On the other hand, highly confident structural predictions, together with conservation of key residues, suggest the M3-like collagen binding mechanism is shared by SDSE and SESZ M proteins (*Figure 6—figure supplement 1*). This has previously been experimentally validated for the FOG (or Stg11) protein of SDSE, which was found to bind to collagens I and IV with high affinity (*Dinkla et al., 2007*; *Nitsche et al., 2006*). SDSE is increasingly recognized as a highly prevalent human pathogen closely resembling GAS in terms of virulence traits and pathologies, including invasive infections, post-infection autoimmune sequelae (*Xie et al., 2024*) and biofilm formation (*Tölken et al., 2024*). Collagen binding by M proteins may be therefore a far more common virulence mechanism in human streptococcal infections than the prevalence of *emm*3 GAS alone would suggest. It should be noted, too, that a new *emm*3 variant, *emm*3.93, is currently emerging in the UK and the Netherlands, with a high prevalence in invasive infections (*Davies et al., 2025*).

The interaction between M protein and collagen has implications for several biological functions, including bacterial attachment and biofilm formation. In this study, we investigated biofilm formation due to the high prevalence of *emm*3 strains in necrotizing soft tissue infections (*Bruun et al., 2021*), in which biofilm is a complicating feature (*Siemens et al., 2016*; *Skutlaberg et al., 2022*). Our findings reveal that the M3 protein–collagen interaction promotes biofilm in *emm*3 strains. Notably, for other *emm* types, the presence of collagen appears to have an opposing effect, reducing bacterial attachment and biofilm in vitro. This type-specific difference is further supported by observations within infected tissue models where *emm*3 GAS showed a more pronounced colocalization with collagen fibers, compared to *emm*1 GAS. These findings support the importance of collagen for *emm*3 biofilm development in the tissue setting, while other GAS *emm* types likely utilize other mechanisms. Further studies are warranted to explore the underlying mechanisms of biofilm formation in streptococcal tissue infections, including not only GAS but also SDSE strains.

## Materials and methods

### Key resources table

| Reagent type (species) or resource | Designation | Source or reference | Identifiers | Additional information |
| --- | --- | --- | --- | --- |
| Strain (*Streptococcus pyogenes*) | 2006, 2028, 5004 | *Siemens et al., 2016* | | Clinical isolates |

*Continued on next page*

*Continued*

| Reagent type (species) or resource | Designation | Source or reference | Identifiers | Additional information |
|---|---|---|---|---|
| Strain (*S. pyogenes*) | 5262 and 8003 | Dr. Donald E. Low, *Kaul et al., 1997*; *Johansson et al., 2008* | | Clinical isolates |
| Strain, strain background (*Escherichia coli*) | BL21(DE3) | Sigma-Aldrich | CMC0016 | Chemically competent cells |
| Cell line (*Homo sapiens*) | Dermal fibroblast (normal, adult) | *Siemens et al., 2016*; *Bergsten et al., 2021*; *Siemens et al., 2015* | NHDF | Isolated from skin biopsies of healthy donors |
| Cell line (*H. sapiens*) | Keratinocytes | Dr. J. Rheinwald, Cell Culture Core of the Harvard Skin Disease Research Centre, Boston, MA, USA, *Siemens et al., 2016*; *Bergsten et al., 2021*; *Siemens et al., 2015* | N/TERT-1 | |
| Antibody | Anti-M3-HVR, rabbit polyclonal | Prof. Gunnar Lindahl, Lund University | | Recognizes HVR of M3 (1:200) |
| Antibody | Anti-M3-BCW, rabbit polyclonal | Prof. Gunnar Lindahl, Lund University | | Recognizes BCW region of M3 (1:2000) |
| Antibody | Anti-rabbit IgG Alexa Fluor 488 conjugate, donkey polyclonal | Abcam | RRID:AB_2768318 | Secondary for M3 staining (1:500) |
| Antibody | Anti-*Streptococcus group* A antibody, goat polyclonal | Abcam | RRID:AB_778136 | Primary for GAS staining (2 µg/mL) |
| Antibody | Anti-collagen IV antibody (COL-94), mouse monoclonal | Abcam | RRID:AB_869201 | Primary for collagen staining (1:400) |
| Antibody | Anti-goat IgG Alexa Fluor 488 conjugate, donkey polyclonal | Abcam | RRID:AB_2687506 | Secondary for GAS staining (1:500) |
| Antibody | Anti-mouse IgG Alexa Fluor 546 conjugate, donkey polyclonal | Invitrogen | RRID:AB_11180613 | Secondary for collagen staining (1:500) |
| Recombinant DNA reagent | pGEX6P-1- M3 (plasmid) | Dr. Susanne Talay, *Dinkla et al., 2003* | | N-terminally GST-fused M3 |
| Recombinant DNA reagent | pEHISTEV-M3NTD (plasmid) | This paper, *Liu and Naismith, 2009* | | N-terminally His6-tagged, TEV cleavable M3-NTD |
| Sequence-based reagent | M3_F | This paper | PCR primer | GCTAGCCATGGATGCTAGGAGTGTTAATGG |
| Sequence-based reagent | M3_R | This paper | PCR primer | CTAGGGATCCCTAGCAGTCCTGATATTCCTTTTC |
| Sequence-based reagent | M3_Y96A_F | This paper | PCR primer | GACAAAAGGCTGAAGCGCTAAAAGGCC |
| Sequence-based reagent | M3_Y96A_R | This paper | PCR primer | GGCCTTTTAGCGCTTCAGCCTTTTGTC |
| Sequence-based reagent | M3_Y96F_F | This paper | PCR primer | GACAAAAGGCTGAATTTCTAAAAGGCC |
| Sequence-based reagent | M3_Y96F_R | This paper | PCR primer | GGCCTTTTAGAAATTCAGCCTTTTGTC |
| Sequence-based reagent | M3_W103A_F | This paper | PCR primer | CCTTAATGATGCGGCTGAGAGGC |
| Sequence-based reagent | M3_W103A_R | This paper | PCR primer | CCTTTTAGATATTCAGCCTTTTG |

*Continued on next page*

*Continued*

| Reagent type (species) or resource | Designation | Source or reference | Identifiers | Additional information |
|---|---|---|---|---|
| Sequence-based reagent | M3_W103D_F | This paper | PCR primer | CCTTAATGATGATGCTGAGAGGCTG |
| Sequence-based reagent | M3_W103D_R | This paper | PCR primer | CCTTTTAGATATTCAGCCTTTTG |
| Sequence-based reagent | M3_W103I_F | This paper | PCR primer | CCTTAATGATATTGCTGAGAGGCTG |
| Sequence-based reagent | M3_W103I_R | This paper | PCR primer | CCTTTTAGATATTCAGCCTTTTG |
| Sequence-based reagent | M3_I60M_F | This paper | PCR primer | GTTAAATTAAAAAATGAAATGGAGAACTTGTTAGATC |
| Sequence-based reagent | M3_I60M_R | This paper | PCR primer | GATCTAACAAGTTCTCCATTTCATTTTTTAATTTAAC |
| Sequence-based reagent | M3_I141M_F | This paper | PCR primer | GAACTTAAGGAAAAAATGGACAAAAAGGAAAAGG |
| Sequence-based reagent | M3_I141M_R | This paper | PCR primer | CCTTTTCCTTTTTGTCCATTTTTTCCTTAAGTTC |
| Commercial assay or kit | CLCII and CLCIII | Triple Helical Peptides Ltd. ***Konitsiotis et al., 2008***; ***Howes et al., 2014*** | | Collagen triple helical peptide libraries |
| Software, algorithm | Topspin | Bruker | RRID:SCR_014227 | |
| Software, algorithm | AlphaFold server | Google DeepMind | RRID:SCR_025885 | |
| Software, algorithm | autoPROC | Global Phasing | RRID:SCR_015748 | |
| Software, algorithm | Coot | ***Emsley et al., 2010*** | RRID:SCR_014222 | |
| Software, algorithm | RefMac5 | CCP4, ***Murshudov et al., 2011*** | RRID:SCR_014225 | |
| Software, algorithm | PyMOL | Schrödinger, LLC | RRID:SCR_000305 | |
| Software, algorithm | PHASER | Phenix, ***McCoy et al., 2007*** | RRID:SCR_014219 | |
| Software, algorithm | MolProbity | Phenix, ***McCoy et al., 2007*** | RRID:SCR_014226 | |
| Other | PDBe | European Bioinformatics Institute | RRID:SCR_004312 | Data deposition service for structural coordinates |

## Cloning and site-directed mutagenesis

M3-NTD DNA insert was amplified from a plasmid containing full M3 DNA sequence using primers M3_F and M3_R, digested with NcoI and BamHI and ligated into the pEHISTEV vector (***Liu and Naismith, 2009***), pre-digested with the same enzymes. The final M3-NTD construct, after digestion with TEV protease, comprised 113 amino acids starting with Gly-Ala-Met, an artifact of the purification tag. Site-directed mutagenesis was performed by PCR amplification of pEHISTEV-M3NTD plasmid using mutagenic primers listed in the Key Resources Table. For selenomethionine labeling, two methionine residues were introduced by substituting Ile60 and Ile141 using two sequential rounds of site-directed mutagenesis.

## Recombinant protein production

Recombinant M3 protein was produced as a GST fusion construct as previously described (***Dinkla et al., 2009***) from a pGEX6P-1 vector, a kind gift from Dr. Susanne Talay, in *E. coli* BL21 (DE3) cells. The cells were grown in LB media containing 100 mg/mL ampicillin at 37°C for 3 h after growth to an optical density at 600 nm of 0.6, and induction with 1 mM isopropyl β-D-1-thiogalactopyranoside (IPTG). The fusion protein was purified from bacterial cell lysate using a GSTrap 4B column according to the protocol provided by the manufacturer (Cytiva). The recombinant protein comprised GST fused to a proteolytic cleavage site (not used in this study) and the N-terminus of M3 (UniProt entry A0A0H2UWN1), lacking N-terminal secretion signal, cell wall anchor, and membrane spanning region (residues 42–546).

M3-NTD constructs were overexpressed from pEHISTEV plasmid in *E. coli* soluBL21 cells (Genlantis), in a standard LB medium supplemented with 50 mg/mL kanamycin. An overnight culture was used to inoculate flasks containing 1 L of the growth medium which were incubated at 37°C with shaking. When the optical density at 600 nm was 0.5–0.8 expression was induced by adding IPTG to a final concentration of 1 mM. After 3–4 h of further incubation with shaking, the cells were harvested and frozen. For selenomethionine labeling, the pHT-M3-NTD [Ile60Met, Ile141Met] plasmid was introduced into the methionine auxotroph strain of *E. coli*, B834, which was obtained from Dr. Clarissa Melo Czekster. The cells from the overnight culture were washed twice with M9 minimal media, resuspended in 0.2× the initial volume, and used to inoculate (10 mL per L) of SelenoMet medium (Molecular Dimensions) supplemented with 0.04 mg/L L-selenomethionine (Thermo Fisher). Following induction of expression, the cells were incubated with shaking for 5 h.

Cell pellets were resuspended in 50–100 mL 50 mM Tris-HCl pH 8, 500 mM NaCl lysis buffer supplemented with one cOmplete protease inhibitor tablet (Roche) and 1 mg DNAse I (Merck). The suspension was passed twice through a Cell Disrupter (Constant Systems) at 30 kpsi and the lysate was clarified by centrifugation at 20,000 × *g* for 25 min at 4°C. The supernatant was applied to a HisTrap column (Cytiva) pre-equilibrated in the lysis buffer and the bound protein was washed with 10 column volumes of the wash buffer (50 mM Tris-HCl pH 8, 500 mM NaCl, 20 mM imidazole) and eluted in a linear gradient (20–250 mM imidazole). To remove imidazole and the N-terminal purification tag, the eluate was mixed with TEV protease (produced in-house) at approximately 30:1 stoichiometric ratio and dialyzed overnight at room temperature against phosphate-buffered saline (PBS) supplemented with 1 mM dithiothreitol (DTT). The digested protein solution was passed through the HisTrap column again, with PBS as the mobile phase, and M3-NTD collected in the flow-through. Following concentration in 10 kDa Amicon Ultra centrifugal filters (Millipore), the protein was further purified by size exclusion chromatography using Superdex 75 10/300 column (Cytiva) pre-equilibrated in PBS. Purity and oxidation state were verified by SDS-PAGE. If some monomeric protein was still evident on the gel, the fractions from size exclusion chromatography were left overnight at 4°C. Fully oxidized protein was concentrated, aliquoted, and flash frozen.

## Collagen peptides

The CLC peptides (formerly known as Collagen Toolkits) were obtained as C-terminal amides from Triple Helical Peptides Ltd, Cambridge, UK. They were synthesized on TentaGel R-Ram resin using Fmoc/tBu chemistry, either on an Applied Biosystems Pioneer peptide synthesizer as described previously (*Raynal et al., 2006*), or a CEM Liberty or Liberty Blue microwave-assisted peptide synthesizer. Fractions containing homogeneous product were identified by analytical HPLC on an ACEphenyl300 (5 mm) column, characterized by MALDI-TOF mass spectrometry, pooled, and freeze-dried. In the CLCs, the variable 27-residue primary collagen structure (guest sequence) was flanked by GPC[GPP]$_5$- and -[GPP]$_5$GPC (host) peptides, to ensure stable triple-helical form. In JDM238, the guest sequence is GVOGEA and the host sequences [GPO]$_3$.

## Solid-phase CLC binding assay

CLC peptide solid-phase binding assays were performed following a previously published protocol (*Konitsiotis et al., 2008*; *Howes et al., 2014*). Briefly, CLC peptides, collagen II (positive control), GPP10, and bovine serum albumin (BSA) (negative controls) (10 µg/mL in 0.01 M acetic acid) were immobilized on Immulon2 HB 96-well plates (Nunc, Langenselbold, Germany) overnight at 4°C. All subsequent incubation steps were for 1 h at 25°C. The assay volume was 100 µL per well. Wells were washed three times with adhesion buffer (0.1% [v/v] Tween-20 and 1 mg/mL BSA in PBS) between incubation steps. The wells were blocked with 50 mg/mL BSA in PBS prior to the addition of recombinant GST-M3 at a concentration of 10 µg/mL in adhesion buffer. Bound protein was detected with biotin goat anti-GST antibody (Abcam) at a 1:500 dilution in adhesion buffer, followed by streptavidin-fused horseradish peroxidase (HRP) and 3,3′,5,5′-tetramethylbenzidine liquid substrate system (Sigma), and plates read at 450 nm. Including Tween in the washing steps reduced background signal but did not change the overall outcome for CLC-II. It was omitted for the CLC-III assay.

## NMR

M3-NTD was isotopically labeled by expression in M9 minimal media (0.25 M $Na_2HPO_4$, 0.13 M $KH_2PO_4$, 0.04 M NaCl, 18 mM $^{15}NH_4Cl$, 2.5 mM $MgSO_4$, 0.1 mM $CaCl_2$, 1% [w/v] glucose, 2.125 g/L BDTM Difco Yeast Nitrogen Base without Amino Acids and Ammonium Sulfate [Thermo Fisher] and 50 µg/mL kanamycin) and purified as described above for unlabeled protein. The NMR sample contained 0.4 mM protein in 10 mM phosphate, 50 mM NaCl, pH 6.5, 1.5% (v/v) $D_2O$, without or with addition of 10 mM DTT to generate monomeric M3-NTD. $^1H,^{15}N$ HSQC spectra were recorded on a Bruker Ascend 700 MHz spectrometer equipped with a Prodigy TCI probe and controlled by Bruker Topspin software at 30°C. A standard Bruker pulse sequence with gradients and water flip back pulse (hsqcetfpf3gpsi) was used with 20 transients and spectral resolutions of 14.5 Hz and 41.2 Hz in the direct ($^1H$) and indirect ($^{15}N$) dimension, respectively. Spectra were processed with NMRPipe (*Delaglio et al., 1995*) and visualized with CCPN Analysis 2 (*Vranken et al., 2005*). For 1D 1H NMR spectra, samples were prepared in PBS at 0.2 mM. A standard Bruker pulse sequence with gradients and excitation sculpting for water suppression (zgesgp) was used with 256 transients and a spectral resolution of 0.86 Hz. Spectra were recorded at 25°C and were processed and visualized with Topspin.

## Isothermal titration calorimetry

Freeze-dried CLC peptides were reconstituted by dissolving in PBS (10 mg/mL), incubating the solution at 75°C in a water bath that was left to cool slowly to room temperature overnight. M3-NTD was dialyzed into PBS. Experiments were performed using a MicroCal PEAQ-ITC instrument (Malvern Panalytical) at 25°C in PBS, with the reference power set to 3 µcal/s, stirring speed to 750 rpm and injection speed to 0.5 µL/s. The cell contained CLC peptide at 270 µM (monomer concentration), and the injector syringe contained M3-NTD variants at 1 mM (monomer concentration). The only exception was the M3-NTD Trp103Ile titration, where the cell and syringe solution concentrations were 165 µM and 0.61 mM, respectively. For control experiments determining the heats of dilution, the cell contained PBS buffer only. Titration involved a single injection of 0.4 µL followed by 18 injections of 2 µL syringe solution.

## Protein crystallization and structure determination

Crystallization trials were conducted using the vapor diffusion sitting drop method and several sparse matrix screens. M3-NTD dimer was freshly purified by size exclusion chromatography and concentrated to ~700 µM (monomer concentration). For co-crystallization with JDM238 peptide, the protein was concentrated to 1 mM and then diluted by adding reconstituted peptide so that the final concentration of M3-NTD was 700 µM with triple-helical peptide at 1.1 mM (~1:3 ratio of M3-NTD dimer to triple-helical peptide). The crystals of M3-NTD grew at 20°C within 1–2 weeks in 40–50% 2-methyl-2,4-pentanediol (MPD), and the best one diffracted to 1.92 Å resolution. They were used for streak-seeding the selenomethionine-labeled protein, which resulted in shard-like crystals diffracting to 2.67 Å with the data collected at the selenium K absorption edge. Phasing of the SAD dataset was conducted automatically by the Diamond Light Source (DLS) pipeline FastEP and initial model building was performed using the ARP/wARP software package. The crystals of M3-NTD in complex with the collagen peptide were obtained in 15% polyethylene glycol 10 K, 0.1 M Tris-HCl pH 8.5, 0.29 M $MgSO_4$ and diffracted to 2.32 Å resolution. Indexing, scaling, and merging of data were performed using the autoPROC pipeline at the DLS. The complex structure was solved by molecular replacement with PHASER (*McCoy et al., 2007*) using M3-NTD and collagen peptide (PDB 3P46) structures as search models, with all non-proline residues in the collagen peptide model substituted with alanine. For both structures, iterative model building and refinement was performed using Coot (*Emsley et al., 2010*) and Refmac5 (*Murshudov et al., 2011*). MolProbity (*Chen et al., 2010*) was used for model quality assessment. Figures of protein structure models were produced using PyMOL (Schrödinger, LLC).

## Accession codes

The M3-NTD and M3-NTD/JDM238 protein structures and the data used to derive these have been deposited at the PDBe with accession codes 8p6k and 8p6j, respectively.

## Bacterial strains

GAS strains 2006 (*emm1*), 2028 (*emm3*), and 5004 (*emm28*) are isolates from patients with necrotizing soft tissue infections from the INFECT project (*Siemens et al., 2016*). GAS 5262 and 8003

were provided by Donald E. Low (Mount Sinai Hospital, Toronto, Canada) and used as further *emm3* isolates from patients with necrotizing soft tissue infections (*Kaul et al., 1997*; *Johansson et al., 2008*). All isolates were cultured in either Todd-Hewitt broth supplemented with 1.5% yeast extract or brain heart infusion broth at 37°C under a 5% $CO_2$ atmosphere.

## Biofilm formation assay on a polystyrene surface

Biofilm formation was evaluated by crystal violet staining as described previously, with minor modifications (*Thenmozhi et al., 2011*). Overnight cultures of the bacteria being tested were washed with PBS and diluted to approximately $10^6$ CFU/mL. Bacterial suspensions (100 μL) were seeded into 96-well polystyrene plates (Thermo Fisher Scientific, Waltham, MA, USA). The plates were then incubated for 24 h at 37°C. After incubation, planktonic bacteria were removed by washing with PBS. Plates were then stained with a 0.1% crystal violet solution (Invitrogen, Waltham, MA, USA) for 30 min. Excess crystal violet was removed by washing with PBS, and then the crystal violet that was associated with bacterial cells was eluted with absolute ethanol. The amount of crystal violet (and by association, biofilm) was evaluated by measuring the absorbance at 590 nm using a spectrophotometer. Collagen-coated plates were prepared by incubating plates with 1–10 μg/mL of collagen I overnight at 4°C.

## Confocal microscopic analysis of biofilm formation

The biofilms were formed on an 8-well chamber slide (Lab-Tek, Thermo Fisher Scientific) in the same manner as for the biofilm assay. After removing planktonic bacteria by washing with PBS, the biofilms were fixed with 10% formalin and stained with wheat germ agglutinin (WGA)-Alexa Fluor 488 conjugate (Invitrogen) and Nile red (Invitrogen). The slides were mounted using ProLong Gold Antifade Mountant with DAPI (Invitrogen).

## Biofilm competition assay

Recombinant M3-NTD was added at 10 and 20 μM concentrations in PBS to a collagen-coated 96-well plate (0.5 μg/well) and the plate was incubated for 1 h at room temperature. After removal of the protein solution, 100 μL of strain 2028 culture (ca. $10^3$ CFU/mL) was seeded and incubated for 24 h. Biofilms were stained with crystal violet, as described above.

## Immunofluorescent staining of GAS

A few colonies were suspended in PBS on a glass slide and then fixed in 3.7% formaldehyde in PBS for 15 min. The bacteria were stained with anti-M3 specific antibodies (kindly provided by Prof. Gunnar Lindahl, Lund University), followed by anti-rabbit IgG Alexa Fluor 488 conjugate antibody (Invitrogen). The slides were mounted using ProLong Gold Antifade Mountant with DAPI (Invitrogen).

## Three-dimensional organotypic skin model

The 3D skin models were constructed as described previously (*Siemens et al., 2016*; *Bergsten et al., 2021*; *Siemens et al., 2015*). Briefly, $4.0 \times 10^4$ normal human dermal fibroblasts (NHDF), isolated from skin biopsies of healthy donors (ethics review committee approved, Stockholm, reference 2006/231-31/4) in a collagen matrix (Pure Col, Advanced Biomatrix, Carlsbad, CA, USA) were seeded onto a polymerized cell-free collagen layer in a 6-well filter insert (Corning, Corning, NY, USA). Authentication of NHDF was confirmed by morphological evaluation and immunofluorescence staining. Cultured cells exhibited the characteristic spindle-shaped morphology typical of fibroblasts, and immunophenotyping was performed with lineage-specific markers, demonstrating positive expression of, for example, vimentin. After culturing in Dulbecco's Modified Eagle Medium (DMEM) for 1 week, $1.0 \times 10^6$ human keratinocyte cells N/TERT-1 were seeded onto the NHDF layer. The human keratinocyte cell N/TERT-1 is a gift from Dr. J. Rheinwald and the Cell Culture Core of the Harvard Skin Disease Research Centre, Boston, MA. The cells were from genetically normal individuals and then immortalized. Phenotypical analysis was done by the reference lab mentioned above. The models were cultured in EpiLife medium (Invitrogen) for 3 days and exposed to air for 1 week. For the infection assay, the models were infected with $1 \times 10^6$ CFU of bacteria for 8, 24, or 48 h. Both cell types used here were tested monthly for Mycoplasma (ATCC Universal Mycoplasma detection kit; Cat No: 30-1012K).

## Immunofluorescent staining of patient tissue biopsies and tissue models

Snap-frozen tissue biopsies from patients with necrotizing soft tissue infections caused by either *emm*1 GAS strains (patients 2006 and 2068) or *emm*3 GAS (patients 2028 and 5020) were available from the INFECT patient biobank (*Siemens et al., 2016*). Cryosectioning and staining were performed as previously described (*Siemens et al., 2015*). Briefly, the biopsies and models were embedded in an optimum cutting temperature compound (Sakura, Torrance, CA, USA) and frozen in liquid nitrogen. Cryosectioning (8 μm) was performed using a Leica CM3050 cryostat (Leica, Nußloch, Germany). Sections were then fixed in 3.7% formaldehyde in PBS for 15 min and stained with both anti-GAS (goat polyclonal antibody, Abcam, Cambridge, UK) and anti-collagen IV antibodies (rabbit polyclonal antibody, Abcam), followed by anti-goat IgG-Alexa Fluor 488 antibody for GAS and anti-mouse IgG Alexa Fluor 546 conjugate antibody for collagen IV (Invitrogen). Manders' correlation coefficient was used to quantify the degree of colocalization between fluorophores and assessed with ImageJ V.1.54p using the JACoP plugin V. 2.1.4. (*Bolte and Cordelières, 2006*).

## Statistical analysis

To compare the propensity of specific CLC amino acids to contribute to M3 binding, the $A_{450}$ value for M3 binding to BSA was subtracted from all values obtained from the solid-phase binding assays. This allowed the CLC peptides to be ranked in descending order of $A_{450}$ and divided into three absorbance groups: high, from 0.75 to 0.5; medium, from 0.5 to 0.25; and low, from 0.25 to zero. Data from both CLCs were pooled, and the amino acid abundance in the three groups was compared. For each peptide from CLC-II and CLC-III, the number of occurrences of each amino acid of interest was noted. The non-parametric Kruskal–Wallis test was used to determine whether the amino acid abundance differed between the three groups. For the biofilm experiments, one-way ANOVA with the Tukey's *post hoc* test was used to determine statistically significant differences.

## Acknowledgements

M3-specific antibodies were kindly provided by Professor Gunnar Lindahl, Lund University. We are grateful to Dr. Susanne Talay for the M3-encoding pGEX6P-1 vector. The authors thank Dr. Conny Yu for preparing a batch of M3-NTD for biofilm experiments. We thank the i04 and i24 beamline staff at the Diamond Light Source synchrotron for their help with data collection.

## Additional information

### Funding

| Funder | Grant reference number | Author |
|---|---|---|
| Medical Research Council | MR/N009681/1 | Marta Wojnowska<br>Samir W Hamaia<br>Richard W Farndale<br>Ulrich Schwarz-Linek |
| European Commission | FP6 ASSIST (032390) | Robert M Hagan<br>Anna Norrby-Teglund<br>Ulrich Schwarz-Linek |
| European Commission | FP7 INFECT (305340) | Takeaki Wajima<br>Helena Bergsten<br>Mattias Svensson<br>Oddvar Oppegaard<br>Steinar Skrede<br>Per Arnell<br>Ole Hyldegaard<br>Anna Norrby-Teglund |

| Funder | Grant reference number | Author |
|---|---|---|
| Vetenskapsrådet | 2022-01-202 | Laura Marcela Palma Medina<br>Mattias Svensson<br>Anna Norrby-Teglund |
| Center for Innovative Medicine | FoUI-975603 | Anna Norrby-Teglund |
| British Heart Foundation | RG/09/003/27122 | Dominique Bihan<br>Richard W Farndale |
| British Heart Foundation | SP/15/7/31561 | Jean-Daniel Malcor<br>Richard W Farndale |
| British Heart Foundation | RG/15/4/31268 | Arkadiusz Bonna<br>Richard W Farndale |

The funders had no role in study design, data collection and interpretation, or the decision to submit the work for publication.

## Author contributions

Marta Wojnowska, Takeaki Wajima, Conceptualization, Formal analysis, Investigation, Methodology, Writing – original draft, Writing – review and editing; Tamas Yelland, Hannes Ludewig, Formal analysis; Robert M Hagan, Formal analysis, Investigation; Olivia F McCurry, Investigation; Grant Watt, Investigation, Writing – review and editing; Samir W Hamaia, Resources, Supervision, Investigation, Methodology; Dominique Bihan, Jean-Daniel Malcor, Arkadiusz Bonna, Resources; Helena Bergsten, Laura Marcela Palma Medina, Formal analysis, Methodology, Writing – review and editing; Mattias Svensson, Conceptualization, Supervision, Project administration, Writing – review and editing; Oddvar Oppegaard, Resources, Writing – review and editing; Steinar Skrede, Per Arnell, Resources, Funding acquisition, Project administration, Writing – review and editing; Ole Hyldegaard, Resources, Investigation, Methodology; Richard W Farndale, Conceptualization, Formal analysis, Supervision, Funding acquisition, Project administration, Writing – review and editing; Anna Norrby-Teglund, Resources, Data curation, Supervision, Funding acquisition, Writing – original draft, Project administration, Writing – review and editing; Ulrich Schwarz-Linek, Conceptualization, Resources, Formal analysis, Supervision, Funding acquisition, Investigation, Methodology, Writing – original draft, Project administration, Writing – review and editing

## Author ORCIDs

Marta Wojnowska http://orcid.org/0000-0001-8954-5127
Takeaki Wajima http://orcid.org/0000-0001-9506-707X
Hannes Ludewig http://orcid.org/0000-0003-1130-1442
Olivia F McCurry http://orcid.org/0009-0002-0372-0137
Jean-Daniel Malcor http://orcid.org/0000-0003-4208-1294
Laura Marcela Palma Medina https://orcid.org/0000-0002-4049-9622
Mattias Svensson https://orcid.org/0000-0003-1695-7934
Anna Norrby-Teglund https://orcid.org/0000-0001-9372-1795
Ulrich Schwarz-Linek https://orcid.org/0000-0003-0526-223X

Reviewer #1 (Public review): https://doi.org/10.7554/eLife.105539.3.sa1
Reviewer #2 (Public review): https://doi.org/10.7554/eLife.105539.3.sa2
Author response https://doi.org/10.7554/eLife.105539.3.sa3

# Additional files

## Supplementary files
MDAR checklist

## Data availability
Diffraction data have been deposited in PDB under accession codes 8p6k and 8p6j.

The following datasets were generated:

| Author(s) | Year | Dataset title | Dataset URL | Database and Identifier |
|---|---|---|---|---|
| Wojnowska M, Schwarz-Linek U | 2023 | Structure of the hypervariable region of *Streptococcus pyogenes* M3 protein | https://doi.org/10.2210/pdb8P6K/pdb | Worldwide Protein Data Bank, 10.2210/pdb8P6K/pdb |
| Wojnowska M, Schwarz-Linek U | 2023 | Structure of the hypervariable region of *Streptococcus pyogenes* M3 protein in complex with a collagen peptide | https://doi.org/10.2210/pdb8P6J/pdb | Worldwide Protein Data Bank, 10.2210/pdb8P6J/pdb |

The following previously published datasets were used:

| Author(s) | Year | Dataset title | Dataset URL | Database and Identifier |
|---|---|---|---|---|
| McEwan PA, Emsley J | 2010 | Integrin binding collagen peptide | https://doi.org/10.2210/pdb3P46/pdb | Worldwide Protein Data Bank, 10.2210/pdb3P46/pdb |
| Boudko SP, Bachinger HP | 2015 | Crystal structure of the type IX collagen NC2 hetero-trimerization domain with a guest fragment a2a1a1 of type I collagen | https://doi.org/10.2210/pdb5CTD/pdb | Worldwide Protein Data Bank, 10.2210/pdb5CTD/pdb |

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
