## [Editor Report · eLife Assessment]

M proteins are essential group A streptococci virulence factors that bind to numerous human proteins; a small subset of M proteins, such as M3, have been reported to bind collagen, which is thought to promote tissue adherence. In this **important** paper, the authors provide a **solid** characterization of M3 interactions with collagen. The work raises significant questions regarding the specificity of the structure and its interactions with different collagens, with implications for the variable actions of M protein collagen interactions on biofilm formation.

---

## [Referee Report · Reviewer #1 (Public review)]

Summary:

Wojnowska et al. report structural and functional studies of the interaction of *Streptococcus pyogenes* M3 protein with collagen. They show through X-ray crystallographic studies that the N-terminal hypervariable region of M3 protein forms a T-like structure, and that the T-like structure binds a three-stranded collagen-mimetic peptide. They indicate that the T-like structure is predicted by AlphaFold3 with moderate confidence level in other M proteins that have sequence similarity to M3 protein and M-like proteins from group C and G streptococci. For some, but not all, of these related M and M-like proteins, AlphaFold3 predicts, with moderate confidence level, complexes similar to the one observed for M3-collagen. Functionally, the authors show that emm3 strains form biofilms with more mass when surfaces are coated with collagen, and this effect can be blocked by an M3 protein fragment that contains the T-structure. They also show the co-occurrence of emm3 strains and collagen in patient biopsies and a skin tissue organoid. Puzzlingly, M1 protein has been reported to bind collagen, but collagen inhibits biofilm in a particular emm1 strain but that same emm1 strain colocalizes with collagen in a patient biopsy sample. The implications of the variable actions of collagen on biofilm formation are not clear.

Strengths:

The paper is well written and the results are presented in a logical fashion.

Weaknesses:

A major limitation of the paper is that it is almost entirely observational and lacks detailed molecular investigation. Insufficient details or controls are provided to establish the robustness of the data.

Comments on revisions:

The authors' response to this reviewer's Major issue #1 is inadequate. Their argument is essentially that if they denature the protein, then there is no activity. This does not address the specificity of the structure or its interactions.

They went only part way to addressing this reviewer's Major issue #2. While Figure 8 - supplement 3 shows 1D NMR spectra for M3 protein (what temperature?), it does not establish that stability is unaltered (to a significant degree).

This reviewer's Major issue #3 is one of the major reasons for considering this study to be observational. This reviewer agrees that structural biology is by its nature observational, but modern standards require validation of structural observations. The authors' response is that a mechanistic investigation involving mutant bacterial strains and validation involving mutated proteins is beyond their scope. Therefore, the study remains observational.

Major issue 4 was addressed suitably, but brings up the problematic point that the emm1 2006 strain colocalizes quite well with collagen in a patient biopsy sample but not in other assays. This calls into question the overall interpretability of the patient biopsy data.

The authors have not provided a point-by-point response. Issues that were indicated to be minor previously were deemed to be minor because this reviewer thought that they could easily be addressed in a revision. It appears that the authors have ignored many of these comments, and these issues are therefore now considered to be major issues. For example, no errors are given for Kd measurements, Table 2 is sloppy and lacks the requested information, negative controls are missing (Figure 10 - figure supplement 1), and there is no indication of how many independent times each experiment was done.

And "C4-binding protein" should be corrected to "C4b-binding protein."

---

## [Referee Report · Reviewer #2 (Public review)]

*Streptococcus pyogenes*, or group A streptococci (GAS) can cause diseases ranging skin and mucosal infections, plasma invasion, and post-infection autoimmune syndromes. M proteins are essential GAS virulence factors that include an N-terminal hypervariable region (HVR). M proteins are known to bind to numerous human proteins; a small subset of M proteins were reported to bind collagen, which is thought to promote tissue adherence. In this paper, authors characterize M3 interactions with collagen and its role in biofilm formation. Specifically, they screened different collagen type II and III variants for full-length M3 protein binding using an ELISA-like method, detecting anti-GST antibody signal. By statistical analysis, hydrophobic amino acids and hydroxyproline found to positively support binding, whereas acidic residues and proline negatively impacted binding. The authors applied X-ray crystallography to determine the structure of the N-terminal domain (42-151 amino acids) of M3 protein (M3-NTD). M3-NTD dimmer (PDB 8P6K) forms a T-shaped structure with three helices (H1, H2, H3), which are stabilized by a hydrophobic core, inter-chain salt bridges and hydrogen bonds on H1, H2 helices, and H3 coiled coil. The conserved Gly113 serves as the turning point between H2 and H3. The M3-NTD is co-crystalized with a 24-residue peptide, JDM238, to determine the structure of M3-collagen binding. The structure (PDB 8P6J) shows that two copies of collagen in parallel bind to H1 and H2 of M3-NTD. Among the residues involved binding, conserved Try96 is shown to play a critical role supported by structure and isothermal titration calorimetry (ITC). The authors also apply a crystal-violet assay and fluorescence microscopy to determine that M3 is involved in collagen type I binding, but not M1 or M28. Tissue biopsy staining indicates that M3 strains co-localize with collagen IV-containing tissue, while M1 strains do not. The authors provide generally compelling evidence to show that GAS M3 protein binds to collagen, and plays a critical role in forming biofilms, which contribute to disease pathology. This is a very well-executed study and a well-written report relevant to understanding GAS pathogenesis and approaches to combatting disease; data are also applicable to emerging human pathogen Streptococcus dysgalactiae. One caveat that was not entirely resolved is if/how different collagen types might impact M3 binding and function. Due to the technical constrains, the in vitro structure and other binding assays use type II collagen whereas in vivo, biofilm formation assays and tissue biopsy staining use type I and IV collagen; it was unclear if this difference is significant. One possibility is that M3 has an unbiased binding to all types of collagens, only the distribution of collagens leads to the finding that M3 binds to type IV (basement membrane) and type I (varies of tissue including skin), rather than type II (cartilage).

Comments on revisions:

We are glad to see that the authors addressed our prior comments on M3 binding to different types of collagens in discussion section; adding a prediction of M3 binding to type I collagen (Figure 8-figure supplement 1B and 1C) is helpful to fill in the gap. Although it would be nice to experimentally fill in the gap by putting all types of collagens into one experiment (For example, like Figure 9A, use different types of human collagens to test biofilm formation; or Figure 10, use different types of human collagens to compete for biofilm formation), this appears to be beyond the scope of this paper. Meanwhile, the changes they have made are constructive.

The authors have addressed the majority of our prior comments.

---

## [Author Response]

The following is the authors’ response to the current reviews.

We thank the reviewers for their comments on the initial submission, which helped us improve and extend the paper. We would like to respond specifically to reviewer #1.

We disagree with the broad criticism of this study as being “almost entirely observational” and lacking “detailed molecular investigation”. We report structures and binding data, show mechanistic detail, identify critical residues and structural features underlying biological activity, and present biologically meaningful data demonstrating a role of the interaction of the M3 protein with collagens. We disagree that insufficient details or controls are included. We agree that our report has limitations, such as an understanding of potential emm1 strain binding to collagen, which might play a role in host tissue colonization, but not in biofilm.

In response to issues raised in the initial review, we conducted several new experiments for the revised manuscript. We believe these strengthen what we report. Firstly, as the reviewer suggested, we conducted a binding experiment where the tertiary fold of M3-NTD was disrupted to confirm the T-shaped fold is indeed required for binding to collagen, as might be expected based on the crystal structure of the complex. To achieve this, we did not, as the reviewer states, use denatured protein in the ITC binding experiment. Instead, we used a monomeric form of M3-NTD, which does not adopt a well-defined tertiary structure, but retains all residues in the context of alpha helices. Secondly, we added more evidence for the importance of structural features (amino acid side chains defining the collagen binding site) by analysing the role of Trp103. Together, we provide clear evidence for the specific role of the T-shaped fold of M3-NTD for collagen binding.

Responding to a constructive criticism by reviewer #1 we characterised M3-NTD mutants to demonstrate conservation of overall structure. NMR is an exquisite tool for this as it is highly sensitive to structural changes. It is not clear why the reviewer suggested we should have measured the stability of the proteins, which is irrelevant here. What matters is that the fold is conserved between mutated variants at the chosen experimental temperature (now added to the Methods section), which NMR demonstrates.

We added errors for the ITC-derived dissociation constants.

In the submitted versions of the paper we did not include the negative control requested by reviewer #1 for experiments shown in Figure 10 - figure supplement 1B. In our view this does not add information supporting our findings. However, we have now added two negative controls, staining of emm1 and emm28 strains. As expected, no reactivity was found with the type-specific M3 HVR antiserum while the M3 BCW antiserum showed weak reactivity, in line with some sequence similarity of the C-terminal regions of M proteins.

Table 2 contains essential information, in line with what generally is shown in crystallographic tables in this journal. All other information can be found in the depositions of our data at the PDB. The structures have been scrutinised and checked by the PDB and passed all quality tests.

We stated how many times experiments were done where appropriate. We now added this information for CLC assays (as given in the previously published protocol, refs. 45, 47). ITC was carried out more than once for optimization but the results of single experiments are shown (as is common practice).

The following is the authors’ response to the original reviews.

Many thanks for assessing our submission. We are grateful for the reviews that have informed a revised version of the paper, which includes additional data and modified text to take into account the reviewers’ comments.

We addressed the major limitation identified by Reviewer #1 by including data to demonstrate that collagen binding is indeed dependent on the T-shaped fold (major issue 1). Reviewer #1 suggested this needs to be done through extensive mutational work. This in our view was neither feasible nor necessary. Instead, we used ITC to measure collagen peptide binding using a monomeric form of M3, which preserves all residues including the ones involved in binding, but cannot form the T-shaped structure. This achieves the same as unravelling the T fold through mutations, but without the risk of aJecting binding through altering residues that are involved in both binding and definition of the T fold. The experiment shows a very weak interaction, confirming the fold of the M3-NTD is required for binding activity.

Reviewer #1 finds the study limited for being “almost entirely observational”. Structural biology is by its nature observational, which is not a limitation but the very purpose of this approach. Our study goes beyond observing structures. In the first version of our paper, we identified a critical residue within a previously mapped binding site, and demonstrated through mutagenesis a causal link between presence of this residue on a tertiary fold and collagen binding activity. However, we agree this analysis could have been strengthened by additional mutagenesis, which we carried out and describe in the revised manuscript. This identifies a second residue that is critical for collagen binding. We firmed up these mutational experiments with a characterisation of mutated forms of M3 by NMR spectroscopy to confirm that these mutations did not aJect the overall fold, addressing major issue no. 2 of reviewer #1. We further demonstrate that the interaction between M3 and collagen is the cause of greatly enhanced biofilm formation as observed in patient biopsies and a tissue model of infection. We show that other streptococci that do not possess a surface protein presenting collagen binding sites like M3 do not form collagen-dependent biofilm. We therefore do not think that criticising our study for being almost entirely observational is valid.

Major issue 3:

We agree with the reviewer that it would be useful to carry out experiments with k.o. and complemented strains. Such experiments go beyond the scope of our study, but might be carried out by us or others in the future. We disagree that emm1 is used “as a negative”. Instead, we established that, in contrast to emm3 strains, emm1 strain biofilm formation is not enhanced by collagen.

We addressed major issue 4 by quantifying colocalizations in the patient biopsies and 3D tissue model experiments.

We thank Reviewer #2 for the thorough analysis of our reported findings. The main criticism here (issue 1) concerns the question of whether binding of emm3 streptococci would diJer to diJerent types of collagen. Our collagen peptide binding assays together with the structural data identify the collagen triple helix as the binding site for M3. While collagen types diJer in their distribution, functions and morphology in diJerent tissues, they all have in common triple-helical (COL) regions with high sequence similarity that are non-specifically recognised by M3. Therefore, our data in conjunction with the body of published work showing binding to M3 to collagens I, II, III and IV suggest it is highly likely that emm3 streptococci will indeed bind to all types of collagen in the same manner. We added a statement to the manuscript to make this point more clearly. We also added a prediction of a complex between M3 and a collagen I triple-helical peptide, which supports the idea of conserved binding mechanism for all collagen types. Whether this means all collagen types in the various tissues where they occur are targeted by emm3 streptococci is a very interesting question, however one that goes beyond the scope of our study.

Minor issues identified by the reviewers were addressed through changes in the text and addition of figures.

Summary of changes:

(1) Two new authors have been added due to inclusion of additional data and analysis.

(2) New experimental data included in section "M3-NTD harbors the collagen binding site".

(3) Figure 3 panels A and B assigned and swapped.

(4) Figure 4 changed to include new data and move mutant M3-NTD ITC graphs to supplement.

(5) Table 2 corrected and amended.

(6) AlphaFold3 quality parameters ipTM and pTM added to all figures showing predicted structures.

(7) New supplementary figure added showing crystal packing of M3-NTD/collagen peptide complex.

(8) Figure supplement of predicted M-protein/collagen peptide complexes includes new panel for a type I collagen peptide bound to M3.

(9) New figure supplement showing mutant M3-NTD ITC data.

(10) New figure supplement showing 1D ^1^H NMR spectra of M3-NTD mutants.

(11) Included data for additional M3-NTD mutants assessing role of Trp103 in collagen binding. Text extended to describe and place into context findings from ITC binding studies using these mutants.

(12) Added quantitative analysis of biopsy and tissue model data (Mander's overlap coeJicient).

(13) Corrected and extended table 3 to take into account new primers.

(14) Added experimental details for new NMR and ITC experiments as well as new quantitative image analysis.

(15) Minor adjustments to the text to improve clarity and correct errors.